# Analysis of European ozone trends in the period 1995–2014

Yingying Yan[1,2,3], Andrea Pozzer[1], Narendra Ojha[1], Jintai Lin[2], Jos Lelieveld[1]

[1] Atmospheric Chemistry Department, Max Planck Institute for Chemistry, Mainz, Germany

[2] Laboratory for Climate and Ocean-Atmosphere Studies, Department of Atmospheric and Oceanic Sciences, School of Physics, Peking University, Beijing 100871, China

[3] Department of Atmospheric Sciences, School of Environmental Studies, China University of Geosciences (Wuhan), 430074, Wuhan, China

Email: andrea.pozzer@mpic.de

## Abstract

Surface-based measurements from the EMEP and Airbase networks are used to estimate the changes in surface ozone levels during the 1995–2014 period over Europe. We find significant ozone enhancements (0.20–0.59 $\mu g/m^3/y$ for the annual means; P-value < 0.01 according to an F-test) over the European suburban and urban stations during 1995–2012 based on the Airbase sites. For European background ozone observed at EMEP sites, it is shown that a significantly decreasing trend in the $95^{th}$ percentile ozone concentrations has occurred, especially during noontime (0.9 $\mu g/m^3/y$; P-value < 0.01), while the $5^{th}$ percentile ozone concentrations continued to increase with a trend of 0.3 $\mu g/m^3/y$ (P-value < 0.01) during the study period. With the help of numerical simulations performed with the global chemistry-climate model EMAC, the importance of anthropogenic emissions changes in determining these changes over background sites are investigated. The EMAC model is found to successfully capture the observed temporal variability in mean ozone concentrations, as well as the contrast in the trends of $95^{th}$ and $5^{th}$ percentile ozone over Europe. Sensitivity simulations and statistical analysis show that a decrease in European anthropogenic emissions had contrasting effects on surface ozone trends between the $95^{th}$ and $5^{th}$ percentile levels, and that background ozone levels have been influenced by hemispheric transport, while climate variability generally regulated the inter-annual variations of surface ozone in Europe.

## 1. Introduction

Tropospheric ozone has detrimental effects on human health, and elevated concentrations at the surface are of concern over most of the European region (Hjellbrekke and Solberg, 2002; WHO, 2013; EEA, 2013; Lelieveld et al., 2015). The European Union (EU) Air Quality Directive sets four standards for surface ozone to reduce its impacts on human health and crop yields (http://eur-lex.europa.eu/legal-content/EN/TXT/HTML/?uri=CELEX:32008L0050&from=EN). These standards are: information threshold (1-hour average: 180 $\mu g/m^3$), alert threshold (1-hour average: 240 $\mu g/m^3$), long-term objective (maximum diurnal 8-hour mean: 120 $\mu g/m^3$), and the

target value (long-term objective that should not be exceeded more than 25 days per year, averaged over 3 years). Exceedances are particularly frequent in regions close to high ozone precursor emissions during summer with stagnant meteorological conditions, associated with persistent high temperatures. Since a substantial decrease in precursor concentrations has been achieved in Europe in recent decades, the number of exceedances has declined (Guerreiro et al., 2014), in line with a long-term downward trend of pollution emissions (Colette et al., 2011; Wilson et al., 2012). Further, a number of studies has shown that European ozone levels are on average decreasing in the last 20 years (as example, Jonson et al, 2010). Nevertheless, background ozone changes over Europe are not so clear (Wilson et al., 2012), being sensitive to climate conditions and intercontinental transport of $O_3$ and its precursors, and are significant in view of tropospheric chemistry (Lelieveld and Dentener, 2000; Lawrence and Lelieveld, 2010).

The response of surface ozone to a changing climate, with potentially more frequent heat extremes (Bloomer et al., 2009; Jacob and Winner, 2009; Cooper et al., 2012; Fu et al., 2015; Lin et al., 2015; Simon et al., 2015), and concurrent changes in anthropogenic emissions of precursor gases (Bloomer et al., 2009; Fu et al., 2015; Strode et al., 2015; Yan et al., 2018) may pose a challenge for air quality management. Observation and model-based analyses of ozone trends in responses to climate change (Bloomer et al., 2009), precursor emissions (Bloomer et al., 2009; Lefohn et al., 2010), and long-range transport (Lin et al., 2015) have been conducted for North America (Strode et al., 2015; Lin et al., 2017; Yan et al., 2018), several Asian regions (Brown-Steiner et al., 2015; Lin et al., 2017) and also for Europe (Meleux et al., 2007, Wilson et al., 2012, Jonson et al., 2006). For Europe, the connection between climate and ozone levels has been subject of large number of studies, notably to investigate the effects of climate change on surface ozone levels (Langner et al., 2005; Meleux et al., 2007; Colette et al., 2011; Langner et al., 2012.)

Tropospheric ozone is produced photochemically during daytime, mainly from the photolysis of nitrogen dioxide ($NO_2$), while $NO_2$ levels are strongly influenced by radicals and their precursors, including organic compounds. Due to the complex photo-chemistry involved, the amount of ozone formed responds nonlinearly to changes in precursor emissions and is sensitive to variations in air temperature, radiation and other climatic factors (Fu et al., 2015; Monks et al., 2015; Coates et al., 2016). Ozone can be destroyed via reaction with NO (i.e., ozone titration) especially during nighttime, and thus a reduction in $NO_x$ emissions could result in more ozone (Jhun et al., 2014; Yan et al., 2018). Previous studies of European ozone have focused on daytime or diurnal mean ozone with little attention paid to the daytime-nighttime contrast in ozone changes (Colette et al., 2011; Wilson et al., 2012; Guerreiro et al., 2014).

Our work contrasts the trends of the monthly 5[th] and 95[th] percentile European background ozone levels at hourly levels over the period 1995–2014, based on the hourly ozone measurements from the EMEP network. Additionally, numerical simulations from the global chemistry-climate model ECHAM5/MESSy (EMAC) are conducted to evaluate the model's ability in capturing ozone trends over Europe and to investigate the underlying importance of the meteorology and emission changes for the observed ozone trends.

The manuscript is organized as follows: the observational dataset, model simulations and analysis
methods are described in Section 2. In Section 3, the average linear trends for the European
domain are estimated and analyzed separately for the monthly, seasonal and annual $5^{th}$, $50^{th}$, and
$95^{th}$ percentiles of the observed surface ozone concentrations. We then compare the observed
ozone trends and variability to results of the atmospheric chemistry – general circulation model
EMAC. To investigate the effects of anthropogenic emissions and climate variability on observed
European ozone changes, we conduct a sensitivity simulation with constant emissions and
statistical analysis with the ERA-Interim 2-meter temperature data in Section 4. Followed by the
conclusions in Section 5.
**2. Methods and Data**
**2.1 Ozone measurements**
The hourly ground-level ozone measurements over 1995–2014 have been obtained from the
Chemical Coordination Centre of *European Monitoring and Evaluation Programme (*EMEP)
network (http://www.nilu.no/projects/ccc/emepdata.html). Table 1 shows the number of
measurement sites (varies from 113 to 137) and the percentage of missing hourly data in each
year. Fig. 1 further shows the site distribution. Since many of the stations are not operating
continuously during the study period (Fig. 1), we have included only the sites in the analysis
which fulfill the criteria defined by Cooper et al. (2012). Such data selection criteria are further
applied for the US ozone trends analysis with the EPA-AQS measurements by Yan et al. (2018).
First, we discard the observational days with the valid hourly data less than 66.7% in any daytime
or nighttime. Then, we discard the particular season with less than 60 days containing valid data
in any season. Finally, for any season, we keep the data with valid seasonal mean ozone more
than 15 years during 1995–2014; otherwise we discard the data in all years for the particular
season. Fig. 1 shows the final selected 93 sites satisfying above criteria for the analysis.
As the measurements from EMEP network are carried out under the "Co-operative programme
for monitoring and evaluation of the long-range transmission of air pollutants in Europe", the
monitoring sites are located where there are minimal local influences, and consequently the
observations are representative of relatively large regions (Torseth et al., 2012). In order to
compare the observed ozone levels and changes over urban, suburban and rural sites, we also use
the hourly measurements over 1995–2012 from the European Environment Agency Airbase
system                (https://www.eea.europa.eu/data-and-maps/data/airbase-the-european-air-quality-
database-8#tab-figures-produced; available years: 1973–2012) (Schultz et al., 2017). After
applying the same data selection criteria above, we get a total of 685 sites (289 for urban, 150 for
suburban and 246 for rural).
We calculated the linear trends for the European surface ozone at individual hours, and mean
values for daytime (local time: 07:00–19:00), nighttime (local time: 19:00–07:00) and full days
(24 h). For each daytime or nighttime period, the missing data varies between 6.8 and 34.6%
(Table 1). The monthly 5[th], 50[th] and 95[th] percentile ozone concentrations for each period (per
hour, daytime, nighttime and diurnal) are derived from the lowest, middle and highest 5[th]
percentile hourly ozone mixing ratios of the corresponding period at individual stations in each
month. Averaging over the 93 sites, we then also calculate the trends of different percentile ozone
concentrations over the whole Europe.
To calculate the ozone trends per hour, during daytime, nighttime and per day, we then use the
following statistical trend model (Weatherhead et al., 1998; Yoon and Pozzer, 2014):
$Y_t = \mu + S_t + \omega X_t + N_t$
Where $Y_t$ denotes the monthly time series of ozone, $\mu$ is a constant term representing the offset,
$X_t = t/12$ (with t as month) the number of years in the timeseries, and $\omega$ is the magnitude of the
trend per year. $S_t$ is a seasonal component in the trend estimates. $N_t$ is the residual term of the
interpolation. As the seasonal component does not have much impact on the statistical properties
of the estimates of the other terms in the model, we use the deseasonalized monthly data to
perform the trend analysis with a model of the form:
$Y_t = \mu + \omega X_t + N_t$
Using this formulation the linear trends are also analyzed separately for the observed monthly,
seasonal and annual surface ozone concentration.
The standard deviation of ozone trends over the European stations is calculated with:
$\sigma = \sqrt{\frac{1}{N} \sum_{i=1}^{N} (\omega_i - \alpha)^2}$
where $N$ is the total number of sites, $\omega_i$ is ozone trend at individual sites and $\alpha$ represents the
average ozone trend.
**2.2 ERA-Interim 2-meter temperature data**
To help investigate the underlying effects of climate variability on ozone variations and trends,
we relate the monthly variability of ozone to 2-meter temperature relevant to the European
ground-level meteorology. The 2-meter temperature data is from the reanalysis product ERA-
Interim, provided by the European Centre for Medium Range Weather Forecast (ECMWF)
Public Datasets web interface (http://apps.ecmwf.int/datasets/), covering the data-rich period
from 1979 and continuing in real time (Dee et al., 2011). Compared to the ERA-40, the ERA-
Interim has an improved representation of the hydrological cycle, and stratospheric circulation
(Dee and Uppala, 2009; Dee et al., 2011). The ERA-Interim atmospheric model and reanalysis
system uses cycle 31r2 of ECMWF's Integrated Forecast System (IFS), configured for 60 vertical
levels up to 0.1 hPa. The horizontal-spatial resolution is either in a full T255 spectral resolution
or in the corresponding N128 reduced Gaussian grid (Dee et al., 2011). ERA-Interim assimilates

four analyses per day, at 00, 06, 12 and 18 UTC. ECMWF public website provides a large variety of data in uniform lat/long grids varying from 0.125° to 3°. Out of those, here, we analyze the monthly mean 2-meter temperature data which are archived on the 0.75° latitude by 0.75° longitude grid. Additional information (e.g. on current data availability) is available on the ECMWF website at http://www.ecmwf.int/research/era.

**2.3 Atmospheric chemistry modeling**

The ECHAM5/MESSy Atmospheric Chemistry (EMAC) model has been used to simulate surface ozone for the 1995–2014 periods. The EMAC model applies the second version of the Modular Earth Submodel System (MESSy2) to link multi-institutional computer codes (Jockel et al., 2016). The core atmospheric model is the 5th generation European Centre Hamburg general circulation model (ECHAM5) (Roeckner et al., 2006). EMAC simulated gas-phase tracers as well as aerosols have been extensively evaluated in previous studies (e. g. Pozzer et al., 2007; Pozzer et al., 2012).

In this work, we use the archived RC1SD-base-10a simulation results from the EMAC model conducted by the ESCiMo project (Jockel et al., 2016). The model results were simulated with version 5.3.02 for ECHAM5 and version 2.51 for MESSy. The archived data were obtained with a T42L90MA spatial resolution, i.e., with a T42 spherical representation which is corresponding to a quadratic Gaussian grid with approximately 2.8 latitude by 2.8 longitude, and 90 levels in the vertical, with the top level up to 0.01 hPa. To reproduce the observed meteorology, the method of Newtonian relaxation towards ERA-Interim reanalysis data (Dee et al., 2011) is applied to weakly nudge the dynamics of the general circulation model. Differently from the work of Jöckel et al. (2016), the model was re-run to cover the full period of measurements and also with a 1-hourly temporal resolution for ozone, in order to compare model results with hourly observational data. We also conducted a sensitivity simulation in which the anthropogenic emissions were kept constant (at the 1994 levels), to represent a scenario with fixed emissions throughout the years where observations are available to investigate the effects of emissions on ozone trends.

The chemical mechanism in the simulations considers the basic gas-phase chemistry of ozone, odd nitrogen, methane, alkanes, alkenes and halogens (bromine and chlorine). Here we use the Mainz Isoprene Mechanism (version 1; MIM1) to account for the chemistry of isoprene and additional non-methane hydrocarbons (NMHCs). This mechanism in total includes 310 reactions of 155 species and is included in the submodel MECCA (Jöckel et al., 2010; R. Sander et al., 2011).

Anthropogenic and biomass burning emissions in the model are incorporated as prescribed sources following the Chemistry-Climate Model Initiative (CCMI) recommendations (Eyring et al., 2013), using the MACCity (Monitoring Atmospheric Composition & Climate/City Zero Energy) emission inventory, which includes a seasonal cycle (monthly resolved) for biomass

burning (Diehl et al. 2012) and anthropogenic emissions (Granier et al. 2011). Additionally, the emissions are vertically distributed as described by Pozzer et al. (2009). Since the total NMVOCs (non-methane volatile organic compounds) values for anthropogenic sectors are not provided by the MACCity raw dataset, they are recalculated from the corresponding species (Jockel et al., 2016).

Emissions from natural sources have been prescribed as well, either as monthly resolved or annually constant climatology. The spatial and temporal distributions of biogenic NMHCs are based on Global Emissions InitiAtive (GEIA). In addition, the emissions of terrestrial dimethyl sulfide (DMS), volcanic $SO_2$, halocarbons and ammonia are prescribed mostly based on climatologies. The ocean-to-atmosphere fluxes of DMS, $C_5H_8$, and methanol are calculated by the AIRSEA submodel (Pozzer et al., 2006) following the two-layer model by Liss and Slater (1974). The emissions of soil $NO_x$ (Yienger and Levy, 1995;Ganzeveld et al., 2002) and biogenic isoprene ($C_5H_8$) (Guenther et al., 1995;Ganzeveld et al., 2002) are calculated online using the submodel ONEMIS. The lightning $NO_x$ emissions are calculated with the submodel LNOX (Tost et al., 2007) following the parameterization by Grewe et al. (2001). This scheme links the flash frequency to the thunderstorm cloud updraft velocity. Aerosols are included in the simulation, although their heating rates and surface areas (needed for heterogeneous reactions) are prescribed from an external climatology rather than interactive chemistry. Further details of the model setup on the emissions, physical and chemical processes as well as the model evaluation with observations can be found in Jöckel et al. (2016).

## 3. Results

### 3.1 Ozone trends in EMEP and Airbase measurements

Annual and seasonal mean daytime and nighttime ozone mixing ratios averaged over the EMEP sites and Airbase sites are shown in Fig. 2. Ozone mixing ratios are maximum over the spring-to-summer season and minimum over the fall-to-winter season for different type of station classification. For annual mean ozone, the concentrations both in daytime and at night over rural sites (EMEP sites and Airbase rural sites) are higher than those averaged over the Airbase suburban and urban sites. Although the EMEP (93 sites) ozone and Airbase rural (246 sites) ozone are calculated based on different number of sites, the ozone trends (shown in each panel in Fig. 2) for annual and seasonal means are similar both during daytime and at night. For the Airbase suburban and urban sites, ozone has increased rapidly with the statistically significant growth rates of 0.09–0.83 µg/m$^3$/y, except that a decline rate of -0.19 µg/m$^3$/y (P-value < 0.01) is also visible for suburban summer ozone during 1995–2012. These suburban and urban ozone enhancements (0.20–0.59 µg/m$^3$/y for annual means; P-value < 0.01) are contrast to the slight rural ozone decrease (-0.09 – -0.02 µg/m$^3$/y for annual means; with an increasing trend for winter ozone and a decreasing trend for summer ozone). As the EMAC model version used here is on a coarse resolution, which is not suitable to investigate the observed contrast ozone trends among

the urban, suburban and rural stations, we focus on the analysis of ozone levels and changes over the regional background areas monitored by EMEP network in the following results.

Fig. 3 shows the trends in ozone concentrations (monthly mean, $5^{th}$, $50^{th}$ and $95^{th}$ percentile) over EMEP sites during the 1995–2014 period, for each hour of the day. While the average ozone concentrations (and $50^{th}$ percentiles) do not show significant trends, the $5^{th}$ and $95^{th}$ percentile ozone show significant trends with a clear diel cycle. The $95^{th}$ percentile ozone shows a decreasing trend over Europe during the 1995–2014 period with the trend being most pronounced ($-0.9 \pm 0.5$ μg/m$^3$/y; P-value < 0.01) during midday (1100-1500 h). $95^{th}$ percentile ozone concentrations also show a decreasing trend during the night, however the trends are observed to be smaller ($-0.5 \pm 0.35$ μg/m$^3$/y; P-value < 0.01). For the ozone trend of $95^{th}$ percentile at individual station, 84 sites (90%) are characterized by decreasing trend in daytime and 78 sites (84%) at night (Fig. 5 and Fig. S2). Here the standard deviation depicts the variability of the trends among the stations, and therefore reflects the almost homogeneous decrease over entire Europe. Interestingly, in contrast with the $95^{th}$ percentile, the $5^{th}$ percentile ozone over Europe shows an increasing trend especially during midday ($0.3 \pm 0.16$ μg/m$^3$/y; P-value < 0.01). Further, the temporal evolutions of ozone anomalies during the 1995–2014 period are shown for $5^{th}$ and $95^{th}$ percentile in Fig. S1. The $95^{th}$ percentile ozone trend indicates a general decline in the photochemical buildup of ozone during noon hours, with the exception of strongly enhanced ozone during 2003. The inter-annual variability is observed to be very large with ozone anomalies in excess of 35 μg/m$^3$ in 2003 relative to 2014. For $95^{th}$ percentile ozone, the sharp increase by up to 20 μg/m$^3$ in the year 2003 occurred during a strong European heat wave (Section 4.2). The analysis of individual year observations here shows that the increasing trend in the $5^{th}$ percentile ozone is a robust feature with most of the recent years showing stronger noontime build up in ozone as compared to the 1990s. During the study period the variability in noontime ozone anomalies is however lower (~10 μg/m$^3$) in the $5^{th}$ percentile ozone compared to the $95^{th}$ percentile ozone.

Consistently with the results obtained for hourly ozone, when the observational data is reduced to diurnal values, a growth rate of $0.22 \pm 0.15$ μg/m$^3$/y (P-value < 0.01) is calculated for the European mean $5^{th}$ percentile ozone, while a stronger decline rate of $-0.57 \pm 0.34$ μg/m$^3$/y (P-value < 0.01) is estimated for the European mean $95^{th}$ percentile ozone (see Table 2). Hereafter we will mainly focus on trends in the daytime mean, nighttime mean, $5^{th}$ percentile and $95^{th}$ percentile ozone concentrations.

The observed long-term reduction in $95^{th}$ percentile ozone concentrations over Europe concurs with the reduction in anthropogenic emissions of ozone precursors (Fig. S6). Anthropogenic emissions of $NO_x$ and CO over Europe declined by 35% and 58%, respectively, as calculated from the MACCity inventory. Slower rates of ozone reduction during nighttime are suggested to be combined effects of reduced titration due to lower NOx emissions, and an increase in the global background ozone concentrations during this period, probably due to growing precursor

emissions worldwide since 1995, which has been predicted by Lelieveld and Dentener (2000)
based on atmospheric chemistry – transport modeling, and corroborated by satellite observations
(Richter et al., 2005; Krotkov et al., 2016). The effect of anthropogenic emissions is discussed in
more detail in the Section 4.1.
Fig. 4 further shows ozone trends for each month of the year. The slight growth rates in the 5th
percentile ozone are approximately equally distributed at the level of $0.1 \pm 0.12$ µg/m$^3$/y (P-
value > 0.05), probably due to the absence of ozone diurnal cycle, affected by $NO_x$ anthropogenic
emissions, for 5th percentile especially in winter. Conversely, the monthly trends for the 95th
percentile ozone are negative with a most rapid decrease rate of $-1.67 \pm 0.4$ µg/m$^3$/y (P-value <
0.01) in August. For the 50th percentiles (mean) the seasonal cycle of ozone trends decline
unevenly from January to August, then pick up in the following months. It leads to the fastest
ozone growth in December when the ozone production is minor due to the relatively lowest solar
UV fluxes and temperatures, and the maximum ozone decline in August, which is the
photochemically most active month in Europe. In December, the 50th (mean) percentile ozone
increases at a rate of $0.41 \pm 0.21$ µg/m$^3$/y ($0.32 \pm 0.09$ µg/m$^3$/y), while a decline rate of $-0.40 \pm$
$0.24$ µg/m$^3$/y ($-0.51 \pm 0.13$ µg/m$^3$/y) is calculated in August.
Table 3 shows the trends in European mean (averaged over the 93 sites) seasonal ozone
concentrations analyzed separately for day- and nighttime. The ozone concentrations show
pronounced differences in trends over the different seasons. The mean surface ozone in summer,
averaged over the selected 93 sites, declines at rates of $-0.32 \pm 0.24$ µg/m$^3$/y and $-0.20 \pm 0.27$
µg/m$^3$/y during day- and nighttime, respectively. It is mainly related to the rapid decline in the
highest levels (95th percentile) of ozone with rates of $-1.10 \pm 0.61$ µg/m$^3$/y (daytime) and $-0.71 \pm$
$0.52$ µg/m$^3$/y (nighttime). Although the 95th percentile ozone in spring declines almost as fast as
during summer, the decrease in spring for the 95th percentile ozone is compensated by the growth
in 5th percentile ozone, leading to much lower decrease rates in spring compared to summer for
the mean ozone concentrations. Finally, in winter ozone grows at a rate of ~0.10 µg/m$^3$/y. This
increase occurs mostly in the lower level (5th percentile) ozone concentrations, with growth rates
of $0.25 \pm 0.15$ µg/m$^3$/y (daytime) and $0.14 \pm 0.22$ µg/m$^3$/y (nighttime).
For the trends in annual mean ozone mixing ratios, a decline in the 95th percentile ozone
(daytime: $-0.81 \pm 0.46$ µg/m$^3$/y; nighttime: $-0.57 \pm 0.36$ µg/m$^3$/y) is observed while an increase in
the 5th percentile ozone ($0.22 \pm 0.17$ and $0.16 \pm 0.17$ µg/m$^3$/y for day- and nighttime,
respectively, is calculated, resulting in statistically not-significant decreasing trends (daytime: -
$0.09 \pm 0.24$; nighttime: $-0.05 \pm 0.23$ µg/m$^3$/y) (Table 3).
Fig. 5 further shows the ozone trends distribution site-by-site over the 93 selected stations for
daytime mean, 5th and 95th percentile ozone during the four seasons. The 95th percentile ozone
trend shows a decline at most of the selected sites, although ozone increases are also visible at
several sites, especially in fall-to-winter. The annual ozone trend averaged over all sites during
daytime (-0.62 µg/m$^3$/y) is nearly twice that during nighttime (-0.35 µg/m$^3$/y, Fig. S2). For the 5$^{th}$
percentile ozone, the annual means have grown over the western and central European sites, in
contrast with declines in ozone at other locations over the northern and southern Europe. These
geographical differences in ozone trends are probably explained by the effects of a general
decrease in European anthropogenic precursor emissions, being partly offset by those of climate
variability (see Sect. 4.2 for discussion of Fig. 11 and Fig. S10). Averaged across all sites, the 5$^{th}$
percentile ozone has slightly grown during day- as well as nighttime. The geographical
differences in ozone trends are most significant in spring with an average growth rate of 0.01
µg/m$^3$/y (Fig. 5). The ozone trends spatial distribution in the daytime (Fig. 5) much resembles
that of the ozone trends in nighttime (Fig. S2) for the mean, 5$^{th}$ percentile as well as 95$^{th}$
percentile ozone.

## 3.2 Ozone exceedance trends

Based on the European directive for ozone concentrations limits, we calculate the number of
exceedances for the information threshold and long-term objective (Fig. 6). Averaged over the
selected 93 sites, the exceedances of the information threshold as well as the long-term objective
have declined at rates of -3.2% and -2.5% per year relative to 1995. The decrease accelerated
after the year 2003, during which a European heat wave raised summer temperatures by 20 to
30% (in degrees Celsius) compared to the seasonal average over a large part of the continent,
extending from northern Spain to the Czech Republic and from Germany to Italy. The variations
in the exceedances are inter-annually consistent with the changes in the annual 95$^{th}$ percentile
ozone, with a significant correlation coefficient of 0.93 for information threshold exceedances
and 0.90 for long-term objective exceedances.

## 3.3 Ozone trends from EMAC simulation

The same analysis performed on the observations has been carried out on the EMAC model
results, i.e., for the same period covered by the observations. To ensure spatiotemporal
consistency with the EMEP data, modeled ozone concentrations are sampled at the times and
locations of the measurements.
Fig. 7 compares the time series of modeled and observed monthly mean ozone over Europe.
Although the model overestimates the measurements with a mean bias of 4.3 µg/m$^3$ over the
1995–2014 period, the simulation results are highly correlated with observed ozone, with a
significant correlation coefficient of 0.91. The high bias may be explained by the coarse grid
resolution of 2.8 degrees that was applied, leading to the artificial dispersion of localized $NO_x$
emissions, which optimizes $NO_x$ concentrations over Europe with respect to chemical $O_3$
formation, also noticed by Joeckel et al (2016). Such overestimation of the observed ozone due to
coarse model horizontal resolution has been reported by Lin et al. (2008) and Yan et al. (2014,
2016). The overestimation after 2010 becomes more evident (mean bias 5.4 µg/m$^3$), mostly due

to the emissions used in the model version used, being prescribed up to the year 2005 and predicted in the subsequent period. The modeled ozone biases are slightly higher (mean bias: 5.2 $\mu g/m^3$ and 6.7 $\mu g/m^3$ for 1995–2014 and 2010–2014, respectively) compared to the observed de-seasonalized time series. Nevertheless, EMAC model can reproduce the observed inter-annual and seasonal variability of ozone, with statistically significant correlation coefficients at most observation sites. For the diurnal, daytime as well as nighttime mean ozone averaged across the 93 sites, the model-observation correlation is 0.84–0.92 (0.62–0.70 for de-seasonalized time series).

Fig. 1 also shows the spatial distribution of observed and modeled mean ozone mixing ratios, as well as the modeled biases for every five years during 1995-2014 over the selected 93 sites. It is shown that for most monitoring stations the model overestimates the observed background ozone concentrations with the bias up to 15 $\mu g/m^3$. Ozone overestimation has been observed also in other EMAC simulations when compared to satellite data (Jöckel et al., 2016). Relatively frequent overestimations (> 10 $\mu g/m^3$) occur over the coastal and marine sites where the coarse model resolution mixes the polluted air over land with cleaner air masses. Underestimation of modeled ozone also occurs over several sites located at the central Europe. These simulated ozone underestimations are probably due to the underestimation of precursor emissions (especially $NO_x$) discussed by Oikonomakis et al. (2017).

The EMAC modeled ozone trends per hour are shown in Fig. 7. The agreement with the observationally estimated trends is good, although the model tends to overestimate the trends by 0.12 $\mu g/m^3/y$, 0.23 $\mu g/m^3/y$, 0.08 $\mu g/m^3/y$, and 0.36 $\mu g/m^3/y$ for the mean, 5[th], 50[th] and 95[th] percentile ozone, respectively. The higher ozone overestimation since 2010 may be the dominant reason for the trend overestimation especially for 95[th] percentile. The measured diurnal cycle of the ozone trends (Fig. 3) is well captured by the EMAC model for the 5[th] and 95[th] percentile ozone concentrations. Consistently, the modeled temporal evolutions (Fig. S3) of annual European 5[th] percentile ozone anomalies are larger compared to the observations (~15 $\mu g/m^3$ versus ~10 $\mu g/m^3$ enhancements during photochemical buildup of ozone at midday hours during 1990–2014), while being smaller for the 95[th] percentile (~21 $\mu g/m^3$ versus ~30 $\mu g/m^3$). Further, the EMAC model reproduces the jump in high level ozone concentrations during the year 2003 that was affected by a major heat wave.

For the diurnal mean values, averaged over Europe, the model produces higher growth rates for the 5[th] percentile ozone and weaker decrease rates for the 95[th] percentile ozone compared to the observed trends (Table 2). For the 50[th] percentile and mean ozone trends averaged over Europe, the model shows statistically insignificant changes, similar to the observed trends (Table 2). Fig. S4 further shows the spatial distribution of the simulated diurnal ozone trends. It corroborates that central Europe experiences the highest growth rate for the averaged (also 50[th] percentile) and 5[th] percentile ozone concentrations, and the strongest reduction for the 95[th] percentile ozone during all seasons.

For the trends per month, the EMAC model reproduces the observed variability with statistically significant correlation coefficients of 0.88–0.90 for the mean, 50th and 95th percentile ozone trends (Fig. 4 and Fig. S5). Seasonally, for the 95th percentile ozone the modeled ozone trends are much weaker than from measurements in all seasons except the autumn (Table 3). The decreased higher level ozone is probably driven by the anthropogenic ozone precursor emission decline over these years, which has been studied in previous work of ozone change drivers and corroborated in Sect. 4.1 with a sensitivity simulation. For the 5th percentile ozone, especially for the daytime period, the increasing trends are enhanced in the model results during all seasons (Table 3). The possible reason for these simulated enhanced ozone trends is the overestimation of the decline of European anthropogenic ozone precursor emissions (decreasing more rapidly than observed) in EMAC.

## 4. Anthropogenic emissions and climate variability

### 4.1 Effects of anthropogenic emissions

A sensitivity simulation is conducted with constant global anthropogenic emissions to test the sensitivity of observed European background ozone to inter-annual variability in climate, by removing the effects of anthropogenic emission changes. Consequently, the decline in European emissions (Fig. S6) is removed from the EMAC model. With constant emissions, the modeled ozone shows a slight increase at the midday hours for the 95th percentile and a slight decrease for the 5th percentile, in contrast to the trends calculated from the control simulation. In the sensitivity simulations no significant trend (less than 0.1 $\mu g/m^3/y$) for any hour of the day is found, and also no contrast in ozone trends between the 5th and 95th percentiles (Fig. 8), which was well reproduced by the control simulation. Therefore, it appears that both the decreases in 95th percentile ozone and the enhancements in 5th percentile ozone are associated with the rapid decline in the precursor gases anthropogenic emissions over Europe, notably of $NO_x$, prescribed by the MACCity inventory (Fig. S6). These results reflect the effectiveness in controlling high-level ozone, but being unsuccessful in controlling the lower level ozone. Evidently, the 35% reduction in $NO_x$ emissions in Europe was not sufficient to achieve substantial reductions in ozone, especially of background levels, which are affected by growing emissions in Asia that are transported hemispherically (Lelieveld and Dentener, 2000; Lawrence and Lelieveld, 2010).

Averaging over the selected 93 sites, we calculate the number of exceedances for the information threshold both in the control and the sensitivity simulation (Fig. 9). In the control simulation, the exceedances of the information threshold have declined at rates of -2.5% per year relative to 1995, slightly smaller than the observed decrease rate of -3.2%. The variations in exceedances are inter-annually consistent with the observations, with a significant correlation coefficient of 0.61. However, in the sensitivity simulation, the decline rate (-0.6%) in the exceedances is much smaller than the rates in the control simulation and in the observations.

By fixing the anthropogenic emissions, ozone trends in each month for the 95[th] percentile ozone show no obvious decline but rather a slight enhancement with growth rates of -0.23 – 0.50 μg/m³/y. For the 5[th] percentile ozone and compared to the control simulation, there is no increase but a slight decrease at a rate of -0.51 – 0.15 μg/m³/y in months of the year (Fig. S7). For the trends in annual mean ozone mixing ratios simulated in the sensitivity simulation, an enhancement in the 95[th] percentile ozone (daytime: 0.16 ± 0.18 μg/m³/y; nighttime: 0.10 ± 0.15 μg/m³/y) is calculated while a decline in the 5[th] percentile ozone (-0.11 ± 0.14 and -0.07 ± 0.12 μg/m³/y for daytime and nighttime, respectively) is estimated, contrasting to but smaller in the absolute value than the trends in the control simulation. This contrast has been also shown in the trends for individual hour of the day between control and sensitivity simulations (Fig. 8). These results show that the effects of decline in anthropogenic emissions on European background ozone change are somewhat offset by the impacts of climate variability. This compensation effect is not only for the high level ozone concentrations, which has been reported by previous studies (Lin et al., 2017), but also for the low level ozone concentrations.

## 4.2 Effects of climate variability

### 4.2.1 Heat wave effects

As discussed in number of studies (e.g., Filleul et al, 2006, Vautard et al, 2005, Garcia-Herrera et a 2010, Vieno et al 2010), the 2003 heat waves caused favorable meteorology for ozone buildup, leading to very high ozone concentrations during the summer period (from July to August). Especially, in August 2003, coinciding with a major heat wave in central and northern Europe, massive forest fires were observed from the Terra and MODIS satellite in many parts of Europe, particularly in the south and most pronounced in Portugal and Spain (Pace et al., 2005; Hodzic et al., 2006, 2007; Solberg et al., 2008). Long-range transport of fire emissions have been found to give rise to significantly elevated air pollution concentration and proved to be contributed to the European ozone peak values in August 2003 (Solberg et al., 2008; Tressol et al., 2008; Ordóñez et al., 2010).

Fig. 10 shows the distribution of the difference in the exceedances between 2003 and averaged over 1995-2002 for the information threshold as well as the long-term objective over individual site. Except for some northern sites, the exceedances in 2003 are much more frequent than the average from 1995 to 2002 over most of the observational sites, especially over central Europe. This exceedance anomaly distribution in 2003 relative to the period of 1995-2002 coincides with the 2-meter temperature anomaly distribution, with a statistically significant correlation up to 0.64 (P-value < 0.01 under a *T*-test; Fig. S8).

### 4.2.2 Effects of inter-annual climate variability

The exceedance anomaly of information threshold and long-term objective during the year 2003
with respect to the 1995-2002 period follows the anomaly in ozone concentrations, in turn
consistent with the temperature anomaly. Fig. 11 shows the correlations between the monthly
mean 2-meter temperature and the monthly mean, 5th and 95th percentile ozone for diurnal,
daytime and nighttime concentrations. Most of these site-by-site correlations are statistically
significant (P-value < 0.05 under a *T*-test; shown as triangles in Fig. 11) with high fraction (66%–
91%) of sites for which significant correlation exist. For each metric (mean and percentiles for
diurnal, daytime and nighttime), it corroborates the high correlations over central Europe with
statistically significant values up to ~0.82 (P-value < 0.01). It indicates that the surface ozone
mixing ratios are highly sensitive to enhanced air temperature, being favorable for photochemical
$O_3$ production, which has been reported by previous studies (Lin et al., 2017; Yan et al., 2018 and
references therein). For different seasons, ozone variations in fall are most closely affected by
temperature (Fig. S9), followed by the spring and summer ozone. The weakest linkage between
ozone and temperature is in winter with few sites for which significant correlation exist
especially for 95th percentile.
In contrast to the positive correlations over central and southern stations, ozone concentrations
over the northern and western sites are negative and significantly correlated with temperature,
associated with statistically insignificant correlations at several sites located in the transition
regions from positive-correlation to negative-correlation (Fig. 11). This may be related to the
influence of the Northern Atlantic Oscillation (NAO; a dominant mode of winter climate
variability in the North Atlantic region including Europe; higher correlations with ozone in winter
shown in Fig. S11), which had an opposite impact on ozone over northern and western compared
to central and southern Europe (Fig. S10). This is because the positive NAO phase is associated
with enhanced pressure gradient between subtropical high pressure center (stronger than usual)
and Icelandic low (deeper than normal). It can result in more and stronger winter storms crossing
the Atlantic Ocean on a more northerly pathway, and consequently lead to warm and wet air in
northern Europe. Compared to the impact of temperature, the effect of NAO on ozone are
relatively modest with much lower correlations (Fig. 11 and Fig. S10). The correlations of less
than 30% of the sites pass the significance test (P-value < 0.05). These results underscore that the
large-scale climate variability affects the inter-annual variability of European background ozone.
In the simulation with constant emissions, however, the modeled ozone fluctuation of annual
European ozone anomalies for individual hours is comparable in magnitude with the results in the
control simulation (Fig. S7). In both simulations, the fluctuation dominates around midday for 5th
(~15 μg/m$^3$ in the base simulation versus ~13 μg/m$^3$ in sensitivity simulation) and 95th (~21
μg/m$^3$ versus ~20 μg/m$^3$) percentile ozone (Fig. S7 and Fig. S3). In addition, the variations in the
exceedances of the information threshold are inter-annually consistent with the observations and
the control simulation, with significant correlation coefficients of 0.54 and 0.56, respectively,
comparable to the correlations between observations and control simulation (Fig. 9). Further
correlations between the European averaged monthly mean 2-meter temperature and the modeled
monthly mean (50$^{th}$), 5$^{th}$ and 95$^{th}$ percentile ozone in the sensitivity simulation are statistically
significant with correlation coefficients of 0.69–0.78 for diurnal, day- and nighttime
concentrations, consistent with the correlations (0.70–0.81) between 2-meter temperature and
simulated European ozone in the control simulation. These results clearly show that the ozone
variations are regulated by climate variations.

**5. Conclusions and outlook**

Based on EMEP observed background ozone in the period 1995–2014, we analyzed the annual
and seasonal trends of the mean, the 5$^{th}$, 50$^{th}$ and 95$^{th}$ percentile of the ozone concentrations at
different temporal distributions, i.e., hourly, diurnal, day- and nighttime. Results show that
although reductions in anthropogenic emissions have lowered the peak ozone concentrations
(sites with statistically significant trends: 91 out of 93 sites; 98%), especially during daytime in
the period 1995–2014, the lower level ozone concentrations have increased (sites with
statistically significant trends: 71 out of 93 sites; 76%) continually since 1995 over Europe. This
leads to insignificant trends in the 50$^{th}$ percentile and mean ozone. Both the 5$^{th}$ and 95$^{th}$ percentile
ozone trends follow a diel cycle with largest trends during periods of strong photochemical
activity. These contrasting ozone trends per hour during the day and at different concentration
levels are well reproduced by the EMAC chemistry-climate model, although the model slightly
overestimates observed ozone at the surface. Furthermore, the numbers of exceedances of the
information threshold and long-term objective have continuously declined during the 20-year
period considered, and the decrease has accelerated since the year 2003.
Sensitivity simulations with constant emissions in the EMAC model, and correlation analysis
between modeled ozone and the ERA-Interim 2-meter temperature help distinguish effects of
climate and anthropogenic emissions on ozone variations and trends. Climate variability
generally regulates the interannual variations of European surface ozone, while the changes in
anthropogenic emissions predominantly contribute to ozone trends. However, it appears that the
negative ozone trend due to European emission controls has been counteracted by a climate
related tendency as well as hemispheric dispersion of pollutants from other regions. We note that
our analysis over 1995–2014 is a timeframe too short for the analysis of climate tendencies
(formally a 30-year period is necessary). Thus, here the climate related variability is mainly
driven by the large-scale processes like NAO and heat wave occurrence, which may be
influenced by climate change.
In contrast to the observed diverse trends of European background ozone, significant ozone
enhancements are found for the annual means (0.20–0.59 μg/m$^3$/y) as well as seasonal means
(0.09–0.83 μg/m$^3$/y), both during daytime and at night over the suburban and urban stations
during 1995–2012 based on the Airbase sites. These increasing trends are interesting and should
be investigated further in view of the continuous decline in European anthropogenic emissions.
**Acknowledgements**
We thank Andries De Vries for discussion. We acknowledge the free use of hourly ozone data
from EMEP network (http://www.nilu.no/projects/ccc/emepdata.html) and ERA-Interim 2 meter
temperature data from the ECMWF.

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

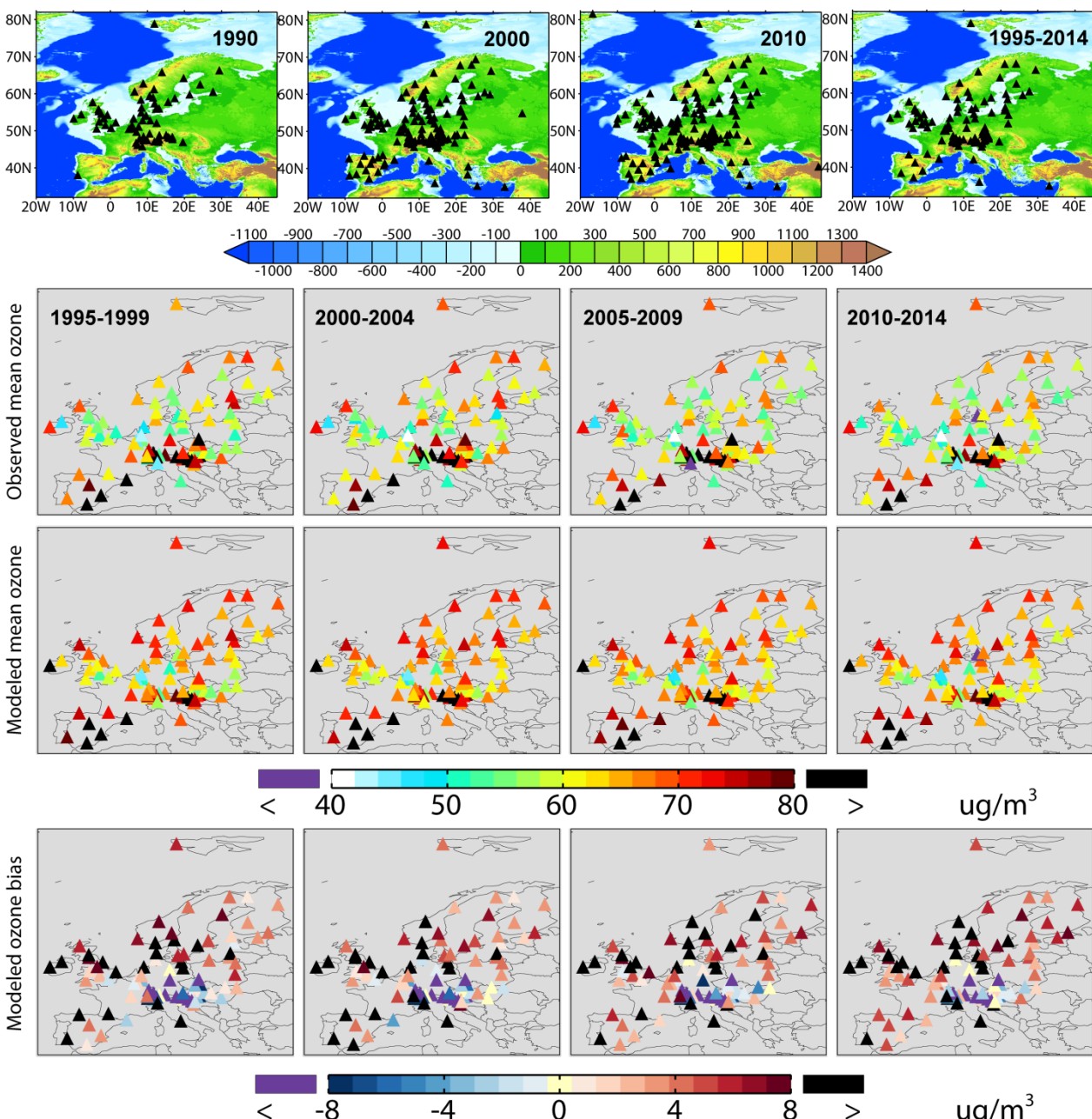

Fig. 1. Site distribution (first row) for the EMEP datasets (1990, 2000, 2010) as well as the selected 93 sites (1995-2014). The overlaid map shows the surface elevation (m) from a 2 min Gridded Global Relief Data (ETOPO2v2) available at NGDC Marine Trackline Geophysical database (http://www.ngdc.noaa.gov/mgg/global/etopo2.html). The observed (second row) and modeled (third row) mean ozone mixing ratios, and also the modeled ozone biases for every five years during 1995-2014 over the selected 93 sites are shown below.


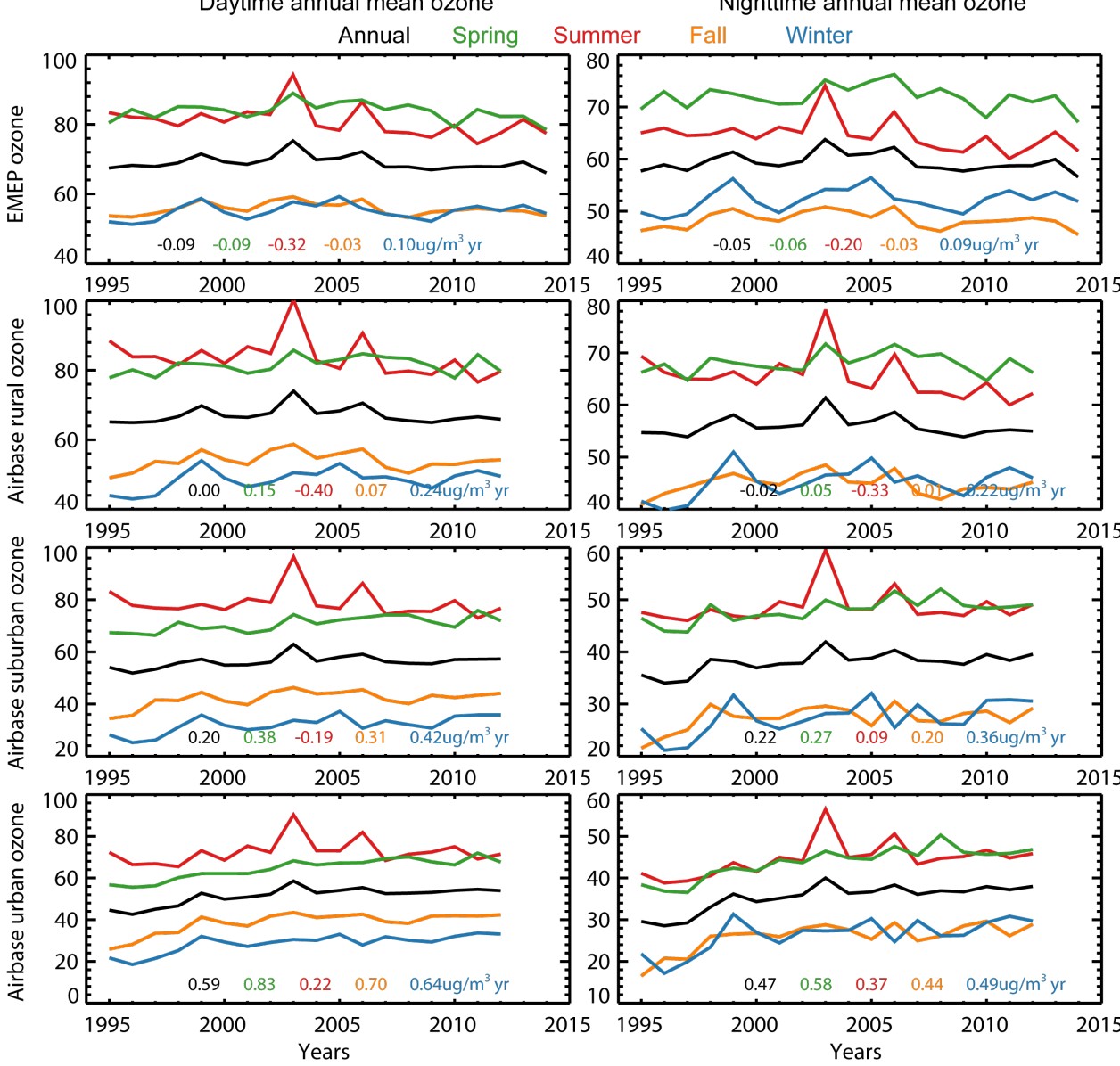

Fig. 2 Annual and seasonal mean daytime and nighttime ozone mixing ratios averaged over the selected sites for
EMEP network (first row) as well as Airbase network (second row for Airbase rural sites; third row for Airbase
suburban sites; forth row for Airbase urban sites). Also shown in each panel are the trends.

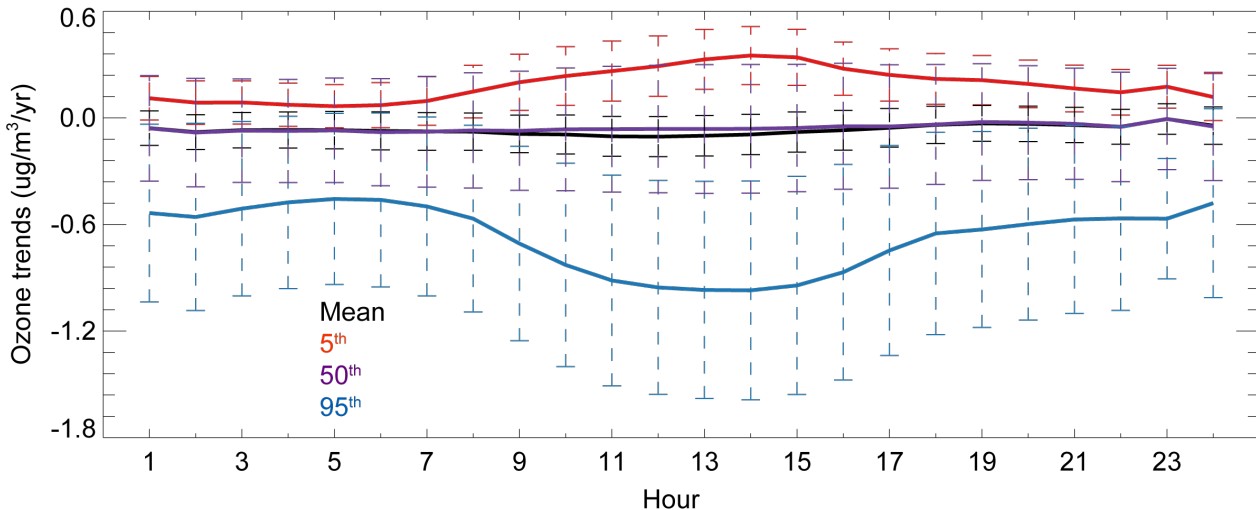

Fig. 3. Trend in the observed surface ozone averaged over Europe, calculated for the selected 93 sites. The black line
shows the 1995–2014 linear trends in the deseasonalized European monthly ozone anomalies for each hour of the
day (local standard time), the red, purple and blue lines depict the observed trend for 5[th], 50[th] and 95[th] percentile
ozone, respectively, and the dashed bars indicate their standard deviations.


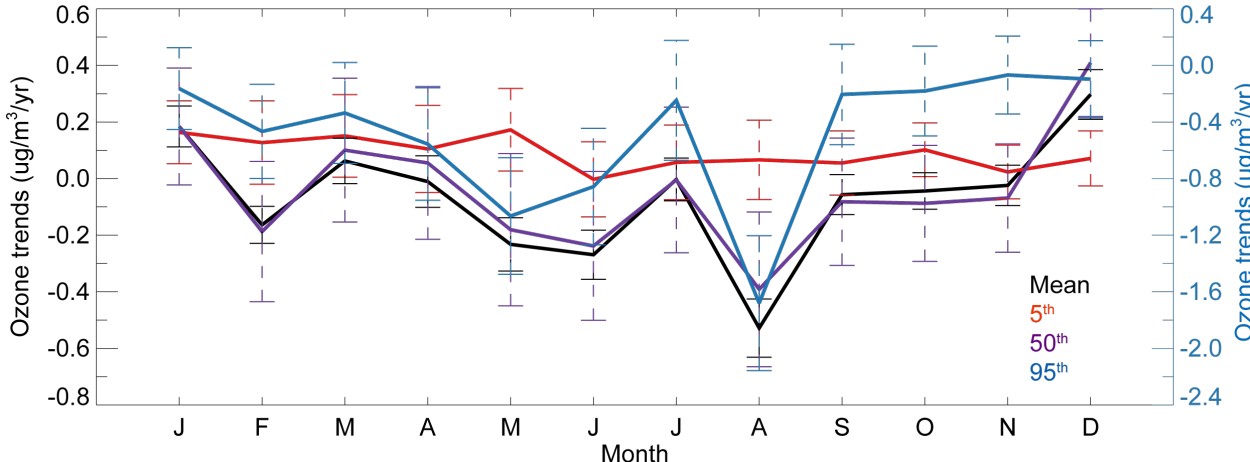

Fig. 4. Monthly trend in the observed surface ozone averaged over Europe for the selected 93 sites. The black line
shows the 1995–2014 linear trends in the European mean ozone for each month of the year, the red, purple and blue
lines depict the observed trend for $5^{th}$, $50^{th}$ and $95^{th}$ percentile ozone, respectively, and the dashed bars indicate their
standard deviations. The left axis is for the trends of mean, $5^{th}$, and $50^{th}$ percentile ozone, while the right axis for the
$95^{th}$ percentile ozone.

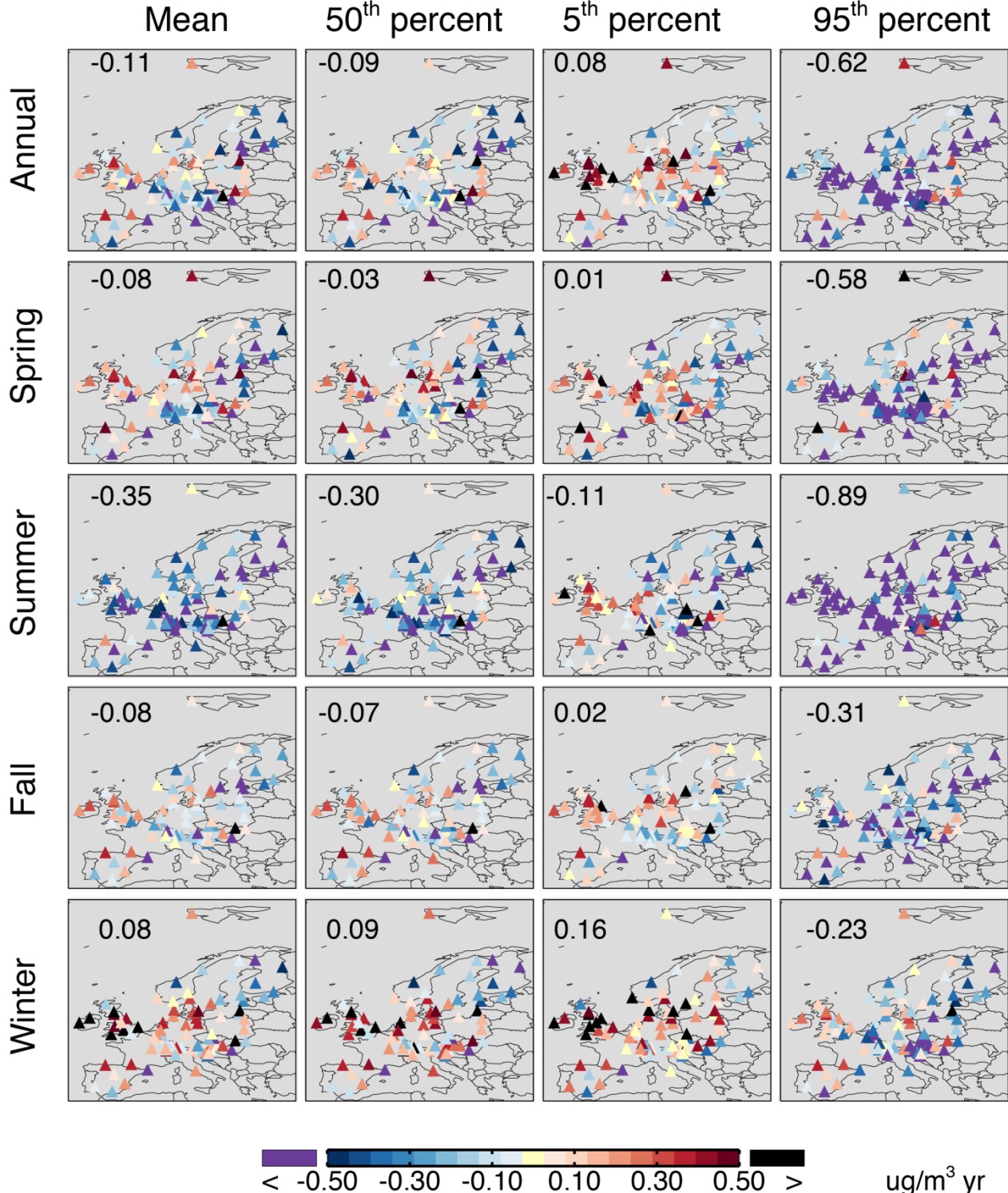

Fig. 5. Spatial distribution of measured daytime ozone trends in μg/m³ across the selected 93 sites for average, 5th, 50th and 95th percentile ozone in annual mean and four seasons. Also shown in each panel are the average trends over all sites.

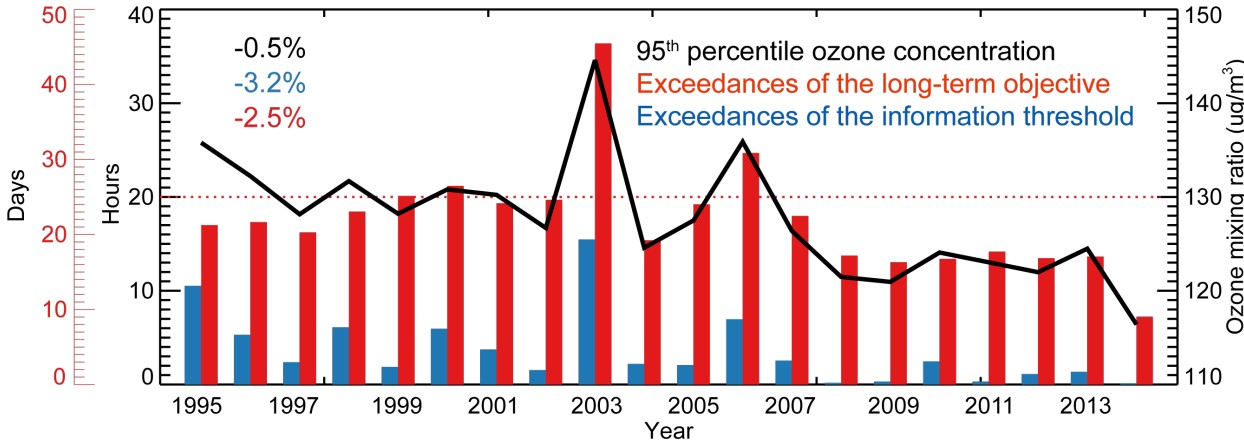

Fig. 6. Annual exceedances of the information threshold (for blue bars, hours should be multiplied by 100, 1-hourly
averages: 180 μg/m³) as well as the long-term objective (red bars, maximum diurnal 8-hourly mean: 120 μg/m³),
compared with the annual 95th percentile ozone concentrations (black line). Red dotted line shows the target value
(long-term objective that should not be exceeded more than 25 days per year, averaged over 3 years).

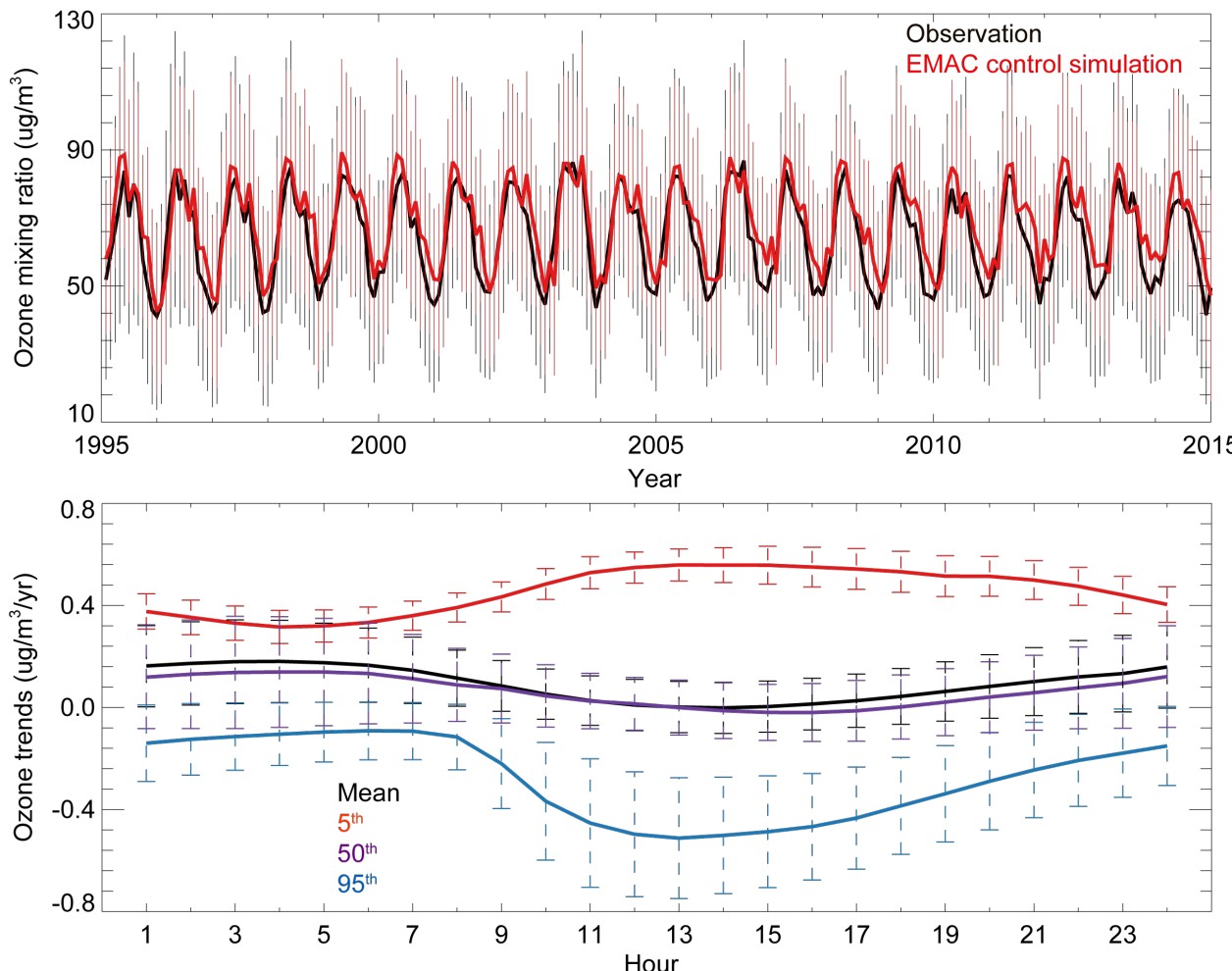

Fig. 7. EMAC modeled ozone in $\mu g/m^3$ over Europe during 1995-2014. Time series of measured (black) and modeled (red) monthly mean ozone over the 93 selected sites (top). Trend in the modeled surface ozone averaged over the selected 93 sites for all hours of the day (local time, bottom). The black line shows the 1995-2014 linear trends in the European mean ozone, the red, purple, and blue lines are the modeled trends for 5th, 50th and 95th percentile ozone, respectively. The dashed bars indicate their standard deviations.

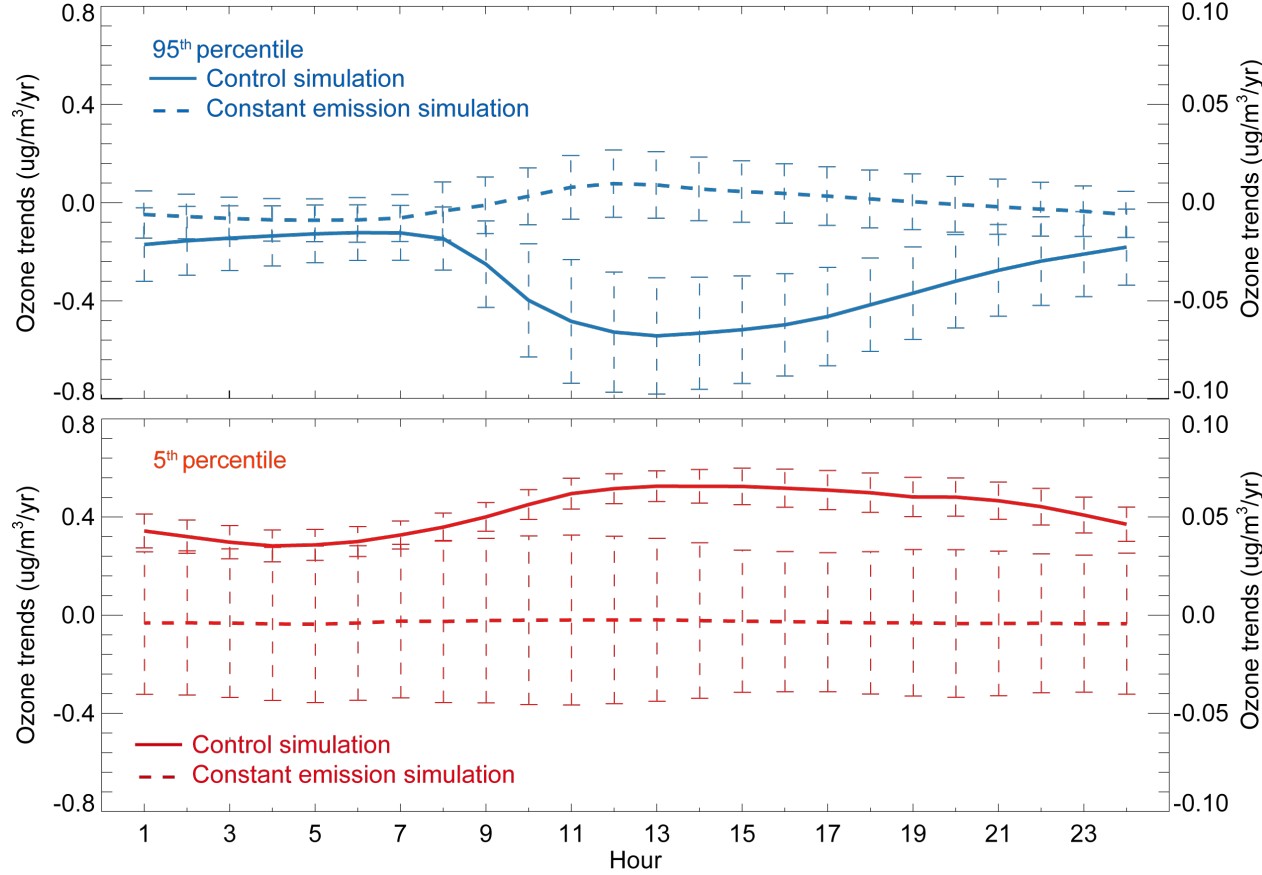

Fig. 8. Modeled trend in the surface ozone averaged over the selected 93 sites for all hours of the day (local time).
The solid lines (left legends) show the 1995-2014 linear trends in the control simulation for 95th (top) and 5th
percentile (bottom) ozone, respectively. The dashed lines (right legends) represent the modeled trends by the
constant emission simulation. The bars indicate their deviations.


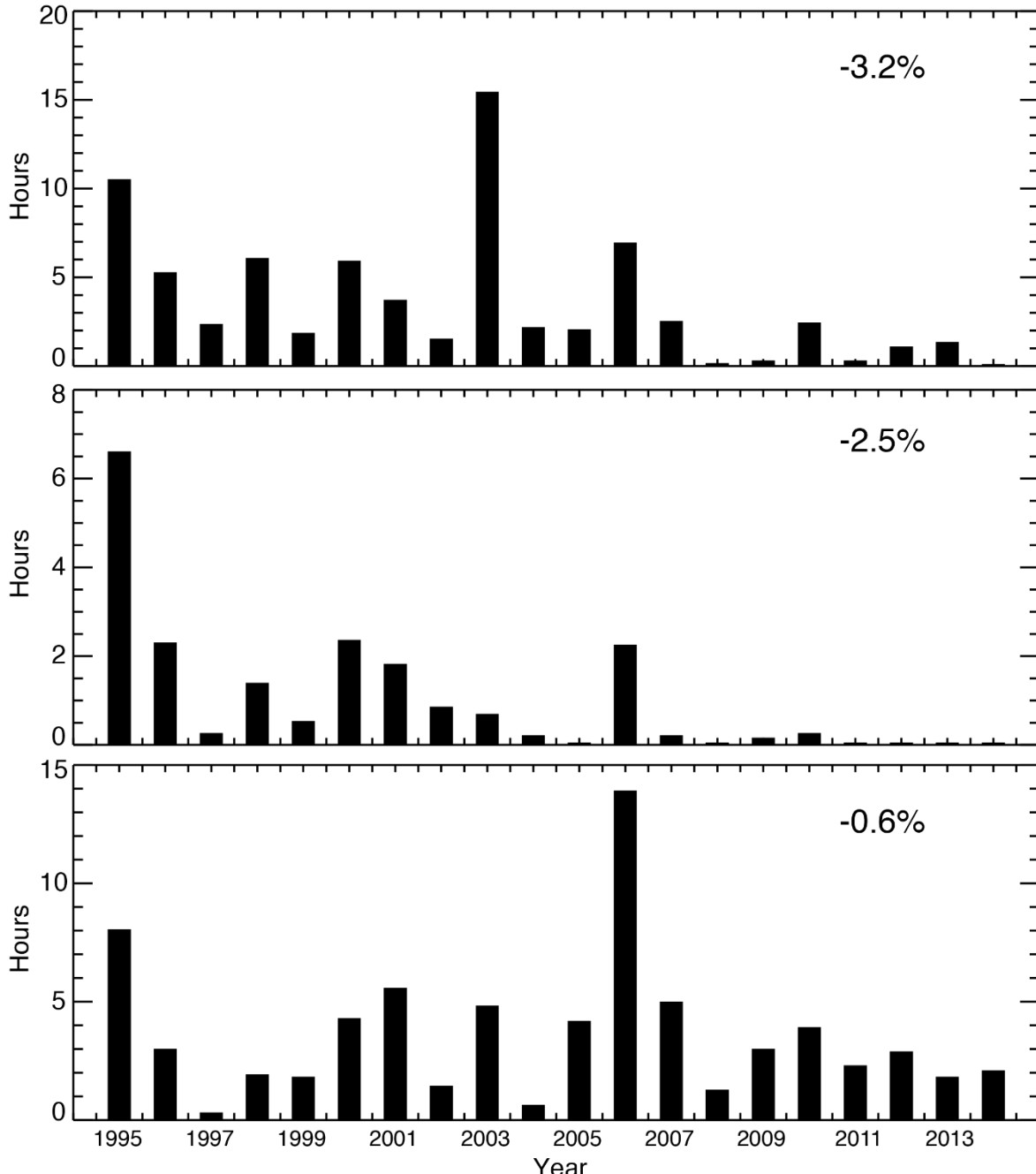

Fig. 9. Annual observed (top) and modeled (middle: control simulation; bottom: constant emission simulation)
exceedances of the information threshold (1-hourly averages: 180 $\mu$g/m$^3$). The hours along the y-axis should be
multiplied by 100.

Fig. 10. Spatial distribution of the exceedance anomalies in 2003, relevant to the averages over 1995-2002 and for the information threshold as well as the long-term objective, in comparison with the 2-meter temperature anomalies in each of the sites.

784

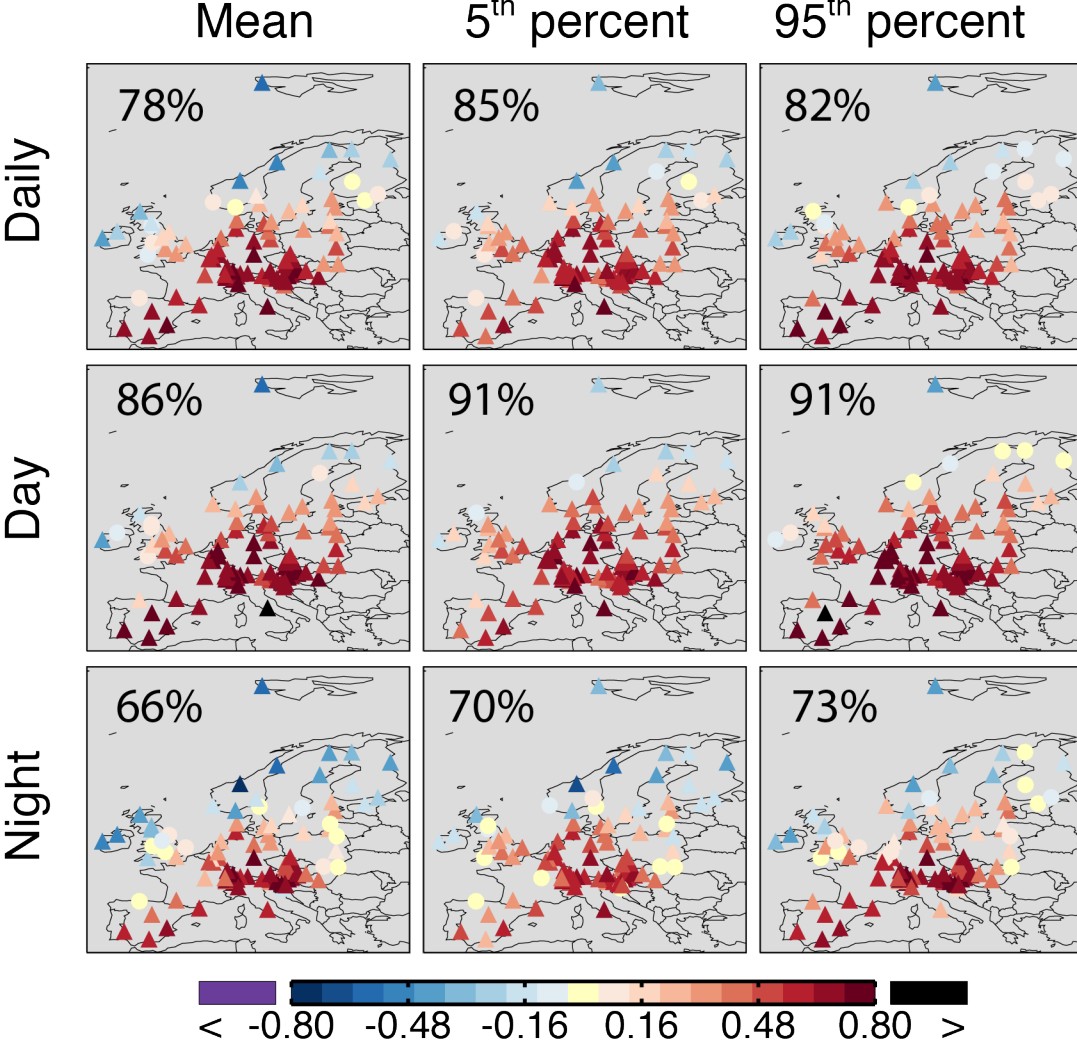

785

Fig. 11. Site-by-site correlations (triangle: P-value < 0.05 under a *T*-test; circular: P-value > 0.05) between the monthly mean 2-meter temperature and monthly mean, 5th and 95th percentile ozone in the daily data, and during daytime as well as nighttime. Also shown in each panel are the fraction of sites for which significant correlation exists.


Table 1. Percentage of missing hourly data in each year in the EMEP station observations.

| Year | Number of sites | Missing data | | |
|---|---|---|---|---|
| | | Whole day | Daytime | Nighttime |
| 1995 | 113 | 32.6% | 30.6% | 34.6% |
| 1996 | 115 | 28.8% | 26.7% | 30.9% |
| 1997 | 121 | 23.9% | 21.6% | 26.2% |
| 1998 | 120 | 18.5% | 16.0% | 21.0% |
| 1999 | 127 | 10.4% | 7.9% | 12.8% |
| 2000 | 132 | 9.8% | 7.2% | 12.3% |
| 2001 | 134 | 11.9% | 9.4% | 14.4% |
| 2002 | 136 | 9.3% | 6.8% | 11.8% |
| 2003 | 137 | 12.1% | 9.8% | 14.4% |
| 2004 | 135 | 10.9% | 8.5% | 13.3% |
| 2005 | 132 | 10.5% | 8.1% | 12.9% |
| 2006 | 130 | 10.6% | 8.1% | 13.1% |
| 2007 | 132 | 9.5% | 7.0% | 12.0% |
| 2008 | 136 | 10.8% | 8.2% | 13.4% |
| 2009 | 134 | 10.6% | 7.8% | 13.3% |
| 2010 | 136 | 15.0% | 12.6% | 17.5% |
| 2011 | 135 | 13.8% | 11.4% | 16.2% |
| 2012 | 136 | 14.1% | 11.8% | 16.4% |
| 2013 | 136 | 19.9% | 17.8% | 22.0% |
| 2014 | 137 | 21.0% | 19.1% | 23.0% |


Table 2. Modeled and observed ozone trends[1] and their standard deviations based on diurnal
average European mean ozone concentrations. The mean, $5^{th}$, $50^{th}$, and $95^{th}$ percentile represent
the monthly statistics of the diurnal averages. The model has been sampled in the same location
of the EMEP stations.

|  | $5^{th}$ percentile | $50^{th}$ percentile | Mean | $95^{th}$ percentile |
|---|---|---|---|---|
| EMEP ($\mu g/m^3$/y) | $0.22^{**} \pm 0.15$ | $-0.05 \pm 0.23$ | $-0.07 \pm 0.21$ | $-0.57^{**} \pm 0.34$ |
| EMAC ($\mu g/m^3$/y) | $0.42^{**} \pm 0.14$ | $0.01 \pm 0.10$ | $0.06 \pm 0.09$ | $-0.23^{**} \pm 0.10$ |

1.   ** P-value < 0.01. * P-value < 0.05 under an *F*-test.

Table 3. Modeled and observed linear trends[1] and their spatial standard deviations of the 1995–
2014 European mean annual and seasonal averaged daytime and nighttime mean as well as their
5th, 50th and 95th percentile ozone concentrations (averaged over the 93 sites).

|  | Seasons | Mean | | 5th percentile | | 50th percentile | | 95th percentile | |
|---|---|---|---|---|---|---|---|---|---|
|  |  | EMEP | EMAC | EMEP | EMAC | EMEP | EMAC | EMEP | EMAC |
| Daytime (µg/m³/y) | Annual | -0.09±0.24 | 0.00±0.06 | 0.22**±0.17 | 0.45**±0.14 | -0.06±0.24 | -0.01±0.06 | -0.81**±0.46 | -0.48**±0.15 |
|  | MAM | -0.09±0.27 | -0.05±0.08 | 0.13±0.24 | 0.52**±0.17 | -0.02±0.27 | -0.02±0.08 | -0.93**±0.53 | -0.49**±0.16 |
|  | JJA | -0.32**±0.24 | -0.10±0.07 | -0.03±0.26 | 0.41**±0.20 | -0.26**±0.24 | -0.09±0.13 | -1.10**±0.61 | -0.54**±0.16 |
|  | SON | -0.03±0.19 | -0.04±0.05 | 0.09±0.14 | 0.36**±0.12 | -0.04±0.20 | -0.02±0.05 | -0.24**±0.25 | -0.44**±0.23 |
|  | DJF | 0.10±0.25 | 0.18**±0.14 | 0.25**±0.15 | 0.39**±0.22 | 0.05±0.27 | 0.15*±0.20 | -0.28**±0.31 | -0.08±0.05 |
| Nighttime (µg/m³/y) | Annual | -0.05±0.23 | 0.12*±0.11 | 0.16*±0.17 | 0.38**±0.19 | -0.05±0.24 | 0.07±0.12 | -0.57**±0.36 | -0.21**±0.10 |
|  | MAM | -0.06±0.29 | 0.08±0.10 | 0.18*±0.23 | 0.23**±0.23 | -0.00±0.29 | 0.04±0.08 | -0.64**±0.43 | -0.20**±0.12 |
|  | JJA | -0.20*±0.27 | 0.06±0.14 | 0.07±0.24 | 0.36**±0.22 | -0.15±0.28 | 0.04±0.14 | -0.71**±0.52 | -0.36**±0.21 |
|  | SON | -0.03±0.21 | 0.06±0.10 | 0.05±0.12 | 0.19**±0.16 | -0.05±0.23 | 0.04±0.11 | -0.21*±0.24 | -0.23**±0.19 |
|  | DJF | 0.09±0.24 | 0.24**±0.18 | 0.14±0.22 | 0.43**±0.27 | 0.06±0.25 | 0.20*±0.25 | -0.24*±0.29 | -0.05±0.06 |

1.  ** P-value < 0.01. * P-value < 0.05 under an $F$-test.