# Peer review of "Analysis of European ozone trends in the period 1995–2014"

_Atmospheric Chemistry and Physics, 2017_

## Referee Comment (RC1) · Anonymous Referee #1 · 12 Jan 2018

The manuscript investigates the European mean, 5th and 95th percentile daily, daytime and nighttime ozone trends between 1995 and 2014, using surface observations from the EMEP network and the EMAC model. The manuscript is well written and organized and the level of the English language is good. It is suitable for publication in ACP after addressing the minor issues I have listed below.

General Comments

Why do the authors use only EMEP stations? There is also other networks available such as AirBase so that there can be an urban background vs. regional investigation of the ozone levels. I am aware that the EMAC model on a coarse resolution is not suitable to investigate the observed trends but limiting it only to the observations would I think increase the value of the paper.

[Figure]

Although the model is well-documented, I think a little more information can be provided for the model properties influencing ozone such as the chemical scheme. Also, more information on how the emissions are used in the model can be useful. Finally, other natural emissions such as dust, sea-salt as well as biomass burning must be explained. The biomass burning during summer time in southern Europe can have significant impacts on o3 levels, which can explain some year-to-year variability.

It would also be interesting to show the spatial evaluation of the MAC model and discuss if there are regions with higher biases than others and why.

Specific Comments

Lines 107-109 is a repetition of lines 94-96.

Section 2.2. lacks motivation for why these analyses will be done for, although it is obvious. I think few lines would improve the flow and readability of the section.

Lines 147-149: Please write here explicitly how the emissions are kept constant? Are they fixed to 1995 or the mean of the period etc. . .?

Line 190: The supplement figure should be referred here.

Line 231: the trends written in the text are slightly different than those on the plots, please double check.

Check the alphabetical order in the Referece list.

Krotkov et al. (2016) is missing in the text.

Langner et al. (2004) is missing in the reference list.

Change Lelieveld et al. (2000) with Lelieveld and Dentener (2010).

Fig. 1. It would be more interesting to see the e.g. annual mean O3 distribution rather than the surface elevation.

Legends should be added to all figures with time series plots.

---

## Referee Comment (RC2) · Anonymous Referee #2 · 18 Jan 2018

**General comments**

This paper by Yingying et al., investigates long-term trend in near-surface ozone in Europe by analysed observations part of the EMEP network. Moreover, it provides some very interesting hints about the different weights that change in European anthropogenic emission and "climate" variability have in determining the observed long-term tendencies.

The paper is well written and within the goal of ACP, the topic is more than relevant. Here, I addressed a few major and minor points that must be considered before final publication in ACP.

MAJOR POINTS

1) One major point that must be carefully addressed by authors is the statistical signif-
icance of the tendencies reported in the paper. As an instance the Mann-Kendall test
must be applied to the different subset of data to verify the actual existence of a "trend".
Otherwise, the authors can only discuss about "tendencies". It is questionable to dis-
cuss and attribute tendencies that are not statistically significant, i.e. not different from
zero. As an instance, statistical significance of tendencies/trends must be indicated in
Table 3.

2) By reading the paper is not clear to me how the authors aggregate data. Are the
monthly percentiles (line 96-98) the average of the corresponding percentiles at each
single station or the percentiles obtained for the whole data set (i.e. by considering all
the ozone data observed at the 93 stations) for each specific month? I think the first
"metric" would be much more robust that the second. . .

3) The analysis concerning the impact of climate variability is promising but it need
more attention: it is not novel that near-surface O3 respond to air-temperature (used
as proxy of meteorological conditions favorable for photochemical production and ac-
cumulation). I would see a more deep discussion (and possibly analysis, see my com-
ment about Fig S9) about the specific processes underlying this "climate variability".
The authors mentioned (and reported by Figure S9) an influence of NAO but without
any specific comments/explanation (I also suggest to discuss possible implication of
NAO to air-mass transport regimes). As suggested by the Referee#1, biomass burn-
ing occurring at continental scale is an issue for near-surface ozone, especially under
heat-wave or dry conditions. A cross-correlation analysis with number or geographi-
cal distribution (burned area) of open forest fire numbers can be useful to assess this
point. For a large subset of year (i.e. since 2000), MODIS data can be used.

4) Line 294-395: the role of China emissions (even if reasonable) is not supported by
data or analysis in this paper. If not strong evidences are added, this statement must
be strongly understated or presented with much more caution. I'm wondering if you
can use EMAC to make sensitivity study on China emission trends by playing with the

MACCity inventory...

SPECIFIC COMMENTS

Line 71: annual "surface" 5th...maybe "surface" ozone concentration?

Line 96: please, better elucidate the aggregation process to obtain the calculated percentiles

Line 170: which is the number sites characterized by negative trend ?

Line 187-190 and Table 3. Are these tendencies/trends obtained by averaging single trends/tendencies at each station or what else? Please specify.

Line 202. Some comments are due to the absence of diurnal cycle for 5th percentile in winter. I would expect a diurnal cycle in NOx anthropogenic emissions that can affect O3 diurnal cycles and subsequently its trends...

Line 214: did you calculate the average of trends or trend of averaged ozone over the whole Europe. In this latter case, you put together sites with very different inhomogeneous in term of ozone variability. As an instance, in summer, ozone is strongly dependent by geographical regions and latitudes...This is also evident by your Figure 4.

Line 233: the annual trend for emep stations here reported are different from those in Table 3. Why?

Line 234-235:please comment these geographical differences and provide possible reasons

Line 237: what do you mean with "regional trend contrast". Contrast in respect to what?

Line 251: why did you investigate the correlation with 95th percentile? What do you want to proof?

Line 275: is the trend overestimation (especially for 95th percentile, i.e. lower decrease

with time) due to the O3 overestimation since 2010?

Line 297: what the reason of the enhanced trends in the 5th percentiles?

Line 359: Figure S9 need to be shown in the main body of paper and it deserve more attention/comments/explanation. As an instance, what the possible impact of NAO variability to transport regimes?

Figure S8-S9: please identify the sites with statistically significant correlation and provide in the paper the fraction of sites for which significant correlation exist for each metric (mean, percentiles) with T and NAO.

Line 352: are these correlation calculated over the 20-yr period? Since NAO effect are strongly dependent by season (see Pausata et al., ACP, 2012), Fig S8 and S9 should be disaggregated as a function of different seasons.

Line 379: it may be useful if the fraction of sites with statistically significant trends is provided.

In the "Conclusion section" it should be stressed that 20-yr is a time frame too short for depict climate tendency (formally a 30 yr period is necessary). I agree that some "large-scale" processes like NAO can influence near-surface O3, thus possible change of these regimes under a changing climate can have serious impact on ozone.

---

## Author Comment (AC1) · 7 Mar 2018

Referee #1

The manuscript investigates the European mean, 5th and 95th percentile daily, daytime and nighttime ozone trends between 1995 and 2014, using surface observations from the EMEP network and the EMAC model. The manuscript is well written and organized and the level of the English language is good. It is suitable for publication in ACP after addressing the minor issues I have listed below.

We thank the reviewer for comments, which have been incorporated to improve the manuscript.

General Comments

1. Why do the authors use only EMEP stations? There is also other networks available such as AirBase so that there can be an urban background vs. regional investigation of the ozone levels. I am aware that the EMAC model on a coarse resolution is not suitable to investigate the observed trends but limiting it only to the observations would I think increase the value of the paper.

In the revised manuscript, we have added the Airbase data to analyze the ozone levels and changes over rural, suburban and urban sites (Fig. 2), and also incorporate these results in conclusion.

In the revised Sect. 2.1, we have added the Airbase data selection: "As the measurements from EMEP network are carried out under the "Co-operative programme for monitoring and evaluation of the long-range transmission of air pollutants in Europe", the monitoring sites are located where there are minimal local influences, and consequently the observations are representative of relatively large regions (Torseth et al., 2012). In order to compare the observed ozone levels and changes over urban, suburban and rural sites, we also use the hourly measurements over 1995–2012 from the European Environment Agency Airbase system (https://www.eea.europa.eu/data-and-maps/data/airbase-the-european-air-quality-database-8#tab-figures-produced; available years: 1973–2012) (Schultz et al., 2017). After applying the same data selection criteria above, we get a total of 685 sites (289 for urban, 150 for suburban and 246 for rural)."

In the revised Sect. 3.1, we have added the Airbase ozone data analysis: "Annual and seasonal mean daytime and nighttime ozone mixing ratios averaged over the EMEP sites and Airbase sites are shown in Fig. 2. Ozone mixing ratios are maximum over the spring-to-summer season and minimum over the fall-to-winter season for different type of station classification. For annual mean ozone, the concentrations both in daytime and at night over rural sites (EMEP sites and Airbase rural sites) are higher than those averaged over the Airbase suburban and urban sites. Although the EMEP (93 sites) ozone and Airbase rural (246 sites) ozone are calculated based on different number

of sites, the ozone trends (shown in each panel in Fig. 2) for annual and seasonal means are similar both during daytime and at night. For the Airbase suburban and urban sites, ozone has increased rapidly with the statistically significant growth rates of 0.09–0.83 $\mu$g/m3/y, except that a decline of -0.19 $\mu$g/m3/y (P-value < 0.01) is also visible for suburban summer ozone during 1995–2012. These suburban and urban ozone enhancements (0.20–0.59 $\mu$g/m3/y for annual means; P-value < 0.01) contrast with the slight rural ozone decrease (-0.09 – -0.02 $\mu$g/m3/y for annual means; with an increasing trend for winter ozone and a decreasing trend for summer ozone). As the EMAC model version used here has a coarse resolution, which is not suitable to investigate the observed contrast ozone trends among the urban, suburban and rural stations, we focus on the analysis of ozone levels and changes over the regional background areas monitored by EMEP network in the following results."

2. Although the model is well-documented, I think a little more information can be provided for the model properties influencing ozone such as the chemical scheme. Also, more information on how the emissions are used in the model can be useful. Finally, other natural emissions such as dust, sea-salt as well as biomass burning must be explained. The biomass burning during summer time in southern Europe can have significant impacts on O3 levels, which can explain some year-to-year variability.

In the revised Sect. 2.3, we have added the information of chemical scheme: "The chemical mechanism in the simulations considers the basic gas-phase chemistry of ozone, odd nitrogen, methane, alkanes, alkenes and halogens (bromine and chlorine). Here we use the Mainz Isoprene Mechanism (version 1; MIM1) to account for the chemistry of isoprene and additional non-methane hydrocarbons (NMHCs). This mechanism in total includes 310 reactions of 155 species and is included in the submodel MECCA (Jöckel et al., 2010; R. Sander et al., 2011)."

Also more emission information has been shown in the revised Sect. 2.3: "Anthropogenic and biomass burning emissions in the model are incorporated as prescribed sources following the Chemistry-Climate Model Initiative (CCMI) recommendations

(Eyring et al., 2013), using the MACCity (Monitoring Atmospheric Composition & Climate/City Zero Energy) emission inventory, which includes a seasonal cycle (monthly resolved) for biomass burning (Diehl et al. 2012) and anthropogenic emissions (Granier et al. 2011). Additionally, the emissions are vertically distributed as described by Pozzer et al. (2009). Since the total NMVOCs (non-methane volatile organic compounds) values for anthropogenic sectors are not provided by the MACCity raw dataset, they are recalculated from the corresponding species (Jockel et al., 2016). Emissions from natural sources have been prescribed as well, either as monthly resolved or annually constant climatology. The spatial and temporal distributions of biogenic NMHCs are based on Global Emissions InitiAtive (GEIA). In addition, the emissions of terrestrial dimethyl sulfide (DMS), volcanic SO2, halocarbons and ammonia are prescribed mostly based on climatologies. The ocean-to-atmosphere fluxes of DMS, C5H8, and methanol are calculated by the AIRSEA submodel (Pozzer et al., 2006) following the two-layer model by Liss and Slater (1974). The emissions of soil NOx (Yienger and Levy, 1995;Ganzeveld et al., 2002) and biogenic isoprene (C5H8) (Guenther et al., 1995;Ganzeveld et al., 2002) are calculated online using the submodel ONEMIS. The lightning NOx emissions are calculated with the submodel LNOX (Tost et al., 2007) following the parameterization by Grewe et al. (2001). This scheme links the flash frequency to the thunderstorm cloud updraft velocity. Aerosols are included in the simulation, although their heating rates and surface areas (needed for heterogeneous reactions) are prescribed from an external climatology rather than interactive chemistry. Further details of the model setup on the emissions, physical and chemical processes as well as the model evaluation with observations can be found in Jöckel et al. (2016)."

3. It would also be interesting to show the spatial evaluation of the MAC model and discuss if there are regions with higher biases than others and why.

In the revised Sect. 3.3, we have shown the spatial evaluation of EMAC modeled ozone with the revised Fig. 1: "Fig. 1 also shows the spatial distribution of observed and modeled mean ozone mixing ratios, as well as the modeled biases for every five

years during 1995-2014 over the selected 93 sites. It is shown that for most monitoring stations the model overestimates the observed background ozone concentrations with the bias up to 15 $\mu$g/m3. Ozone overestimation has been observed also in other EMAC simulations when compared to satellite data (Jöckel et al., 2016). Relatively frequent overestimations (> 10 $\mu$g/m3) occur over the coastal and marine sites where the coarse model resolution mixes the polluted air over land with cleaner air masses. Underestimation of modeled ozone also occurs over several sites located at the central Europe. These simulated ozone underestimations are probably due to the underestimation of precursor emissions (especially NOx) discussed by Oikonomakis et al. (2017)."

Specific Comments

Lines 107-109 is a repetition of lines 94-96.

We have removed the lines 107-109.

Section 2.2. lacks motivation for why these analyses will be done for, although it is obvious. I think few lines would improve the flow and readability of the section.

We have revised the first sentence in Sect. 2.2: "To help investigate the underlying effects of climate variability on ozone variations and trends, we relate the monthly variability of ozone to 2-meter temperature relevant to the European ground-level meteorology."

Lines 147-149: Please write here explicitly how the emissions are kept constant? Are they fixed to 1995 or the mean of the period etc: : :?

This sentence has been revised: "We also conducted a sensitivity simulation in which the anthropogenic emissions were kept constant (at the 1994 levels), to represent a scenario with fixed emissions throughout the years where observations are available to investigate the effects of emissions on ozone trends."

Line 190: The supplement figure should be referred here.

We have added the supplement figure (Fig. S6) reference in the revised sentence.

Line 231: the trends written in the text are slightly different than those on the plots, please double check.

We have modified the trends in the text.

Check the alphabetical order in the Referece list.

We have rearranged the reference list according to the alphabetical order.

Krotkov et al. (2016) is missing in the text.

We have added this reference in the text.

Langner et al. (2004) is missing in the reference list.

We have added Langner et al. (2004) in the reference list.

Change Lelieveld et al. (2000) with Lelieveld and Dentener (2010).

We have added the reference: Lawrence and Lelieveld (2010).

Fig. 1. It would be more interesting to see the e.g. annual mean O3 distribution rather than the surface elevation.

The revised Fig. 1 have added to show mean ozone mixing ratios for every five years during 1995-2014 over the selected 93 sites.

Legends should be added to all figures with time series plots.

We have added legends in the time series plots.

Please also note the supplement to this comment:
https://www.atmos-chem-phys-discuss.net/acp-2017-1077/acp-2017-1077-AC1-supplement.pdf

[Figure]

**Supplement:**

**Referee #1**

The manuscript investigates the European mean, 5th and 95th percentile daily, daytime and nighttime ozone trends between 1995 and 2014, using surface observations from the EMEP network and the EMAC model. The manuscript is well written and organized and the level of the English language is good. It is suitable for publication in ACP after addressing the minor issues I have listed below.

We thank the reviewer for comments, which have been incorporated to improve the manuscript.

**General Comments**

1. Why do the authors use only EMEP stations? There is also other networks available such as AirBase so that there can be an urban background vs. regional investigation of the ozone levels. I am aware that the EMAC model on a coarse resolution is not suitable to investigate the observed trends but limiting it only to the observations would I think increase the value of the paper.

In the revised manuscript, we have added the Airbase data to analyze the ozone levels and changes over rural, suburban and urban sites (Fig. 2), and also incorporate these results in conclusion.

In the revised Sect. 2.1, we have added the Airbase data selection: "As the measurements from EMEP network are carried out under the "Co-operative programme for monitoring and evaluation of the long-range transmission of air pollutants in Europe", the monitoring sites are located where there are minimal local influences, and consequently the observations are representative of relatively large regions (Torseth et al., 2012). In order to compare the observed ozone levels and changes over urban, suburban and rural sites, we also use the hourly measurements over 1995–2012 from the European Environment Agency Airbase system (https://www.eea.europa.eu/data-and-maps/data/airbase-the-european-air-quality-database-8#tab-fi gures-produced; available years: 1973–2012) (Schultz et al., 2017). After applying the same data selection criteria above, we get a total of 685 sites (289 for urban, 150 for suburban and 246 for rural)."

In the revised Sect. 3.1, we have added the Airbase ozone data analysis: "Annual and seasonal mean daytime and nighttime ozone mixing ratios averaged over the EMEP sites and Airbase sites are shown in Fig. 2. Ozone mixing ratios are maximum over the spring-to-summer season and minimum over the fall-to-winter season for different type of station classification. For annual mean ozone, the concentrations both in daytime and at night over rural sites (EMEP sites and Airbase rural sites) are higher than those averaged over the Airbase suburban and urban sites. Although the EMEP (93 sites) ozone and Airbase rural (246 sites) ozone are calculated based on different number of sites, the ozone trends (shown in each panel in Fig. 2) for annual and seasonal means are similar both during daytime and at night. For the Airbase suburban and urban sites, ozone has increased rapidly with the statistically significant growth rates of 0.09–0.83  $\mu$ g/m3/y, except that a decline of -0.19  $\mu$ g/m3/y (P-value < 0.01) is

also visible for suburban summer ozone during 1995–2012. These suburban and urban ozone enhancements (0.20–0.59  $\mu$ g/m3/y for annual means; P-value < 0.01) contrast with the slight rural ozone decrease (-0.09 – -0.02  $\mu$ g/m3/y for annual means; with an increasing trend for winter ozone and a decreasing trend for summer ozone). As the EMAC model version used here has a coarse resolution, which is not suitable to investigate the observed contrast ozone trends among the urban, suburban and rural stations, we focus on the analysis of ozone levels and changes over the regional background areas monitored by EMEP network in the following results."

2. Although the model is well-documented, I think a little more information can be provided for the model properties influencing ozone such as the chemical scheme. Also, more information on how the emissions are used in the model can be useful. Finally, other natural emissions such as dust, sea-salt as well as biomass burning must be explained. The biomass burning during summer time in southern Europe can have significant impacts on O3 levels, which can explain some year-to-year variability.

In the revised Sect. 2.3, we have added the information of chemical scheme: "The chemical mechanism in the simulations considers the basic gas-phase chemistry of ozone, odd nitrogen, methane, alkanes, alkenes and halogens (bromine and chlorine). Here we use the Mainz Isoprene Mechanism (version 1; MIM1) to account for the chemistry of isoprene and additional non-methane hydrocarbons (NMHCs). This mechanism in total includes 310 reactions of 155 species and is included in the submodel MECCA (Jöckel et al., 2010; R. Sander et al., 2011)."

Also more emission information has been shown in the revised Sect. 2.3: "Anthropogenic and biomass burning emissions in the model are incorporated as prescribed sources following the Chemistry-Climate Model Initiative (CCMI) recommendations (Eyring et al., 2013), using the MACCity (Monitoring Atmospheric Composition & Climate/City Zero Energy) emission inventory, which includes a seasonal cycle (monthly resolved) for biomass burning (Diehl et al. 2012) and anthropogenic emissions (Granier et al. 2011). Additionally, the emissions are vertically distributed as described by Pozzer et al. (2009). Since the total NMVOCs (non-methane volatile organic compounds) values for anthropogenic sectors are not provided by the MACCity raw dataset, they are recalculated from the corresponding species (Jockel et al., 2016).

Emissions from natural sources have been prescribed as well, either as monthly resolved or annually constant climatology. The spatial and temporal distributions of biogenic NMHCs are based on Global Emissions InitiAtive (GEIA). In addition, the emissions of terrestrial dimethyl sulfide (DMS), volcanic SO2, halocarbons and ammonia are prescribed mostly based on climatologies. The ocean-to-atmosphere

fluxes of DMS, C5H8, and methanol are calculated by the AIRSEA submodel (Pozzer et al., 2006) following the two-layer model by Liss and Slater (1974). The emissions of soil NOx (Yienger and Levy, 1995;Ganzeveld et al., 2002) and biogenic isoprene (C5H8) (Guenther et al., 1995;Ganzeveld et al., 2002) are calculated online using the submodel ONEMIS. The lightning NOx emissions are calculated with the submodel LNOX (Tost et al., 2007) following the parameterization by Grewe et al. (2001). This scheme links the flash frequency to the thunderstorm cloud updraft velocity. Aerosols are included in the simulation, although their heating rates and surface areas (needed for heterogeneous reactions) are prescribed from an external climatology rather than interactive chemistry. Further details of the model setup on the emissions, physical and chemical processes as well as the model evaluation with observations can be found in Jöckel et al. (2016)."

3. It would also be interesting to show the spatial evaluation of the MAC model and discuss if there are regions with higher biases than others and why.

In the revised Sect. 3.3, we have shown the spatial evaluation of EMAC modeled ozone with the revised Fig. 1: "Fig. 1 also shows the spatial distribution of observed and modeled mean ozone mixing ratios, as well as the modeled biases for every five years during 1995-2014 over the selected 93 sites. It is shown that for most monitoring stations the model overestimates the observed background ozone concentrations with the bias up to 15  $\mu$ g/m3. Ozone overestimation has been observed also in other EMAC simulations when compared to satellite data (Jöckel et al., 2016). Relatively frequent overestimations (> 10  $\mu$ g/m3) occur over the coastal and marine sites where the coarse model resolution mixes the polluted air over land with cleaner air masses. Underestimation of modeled ozone also occurs over several sites located at the central Europe. These simulated ozone underestimations are probably due to the underestimation of precursor emissions (especially NOx) discussed by Oikonomakis et al. (2017)."

**Specific Comments**

Lines 107-109 is a repetition of lines 94-96.

**We have removed the lines 107-109.**

Section 2.2. lacks motivation for why these analyses will be done for, although it is obvious. I think few lines would improve the flow and readability of the section.

We have revised the first sentence in Sect. 2.2: "To help investigate the underlying effects of climate variability on ozone variations and trends, we relate the monthly variability of ozone to 2-meter temperature relevant to the European ground-level meteorology."

Lines 147-149: Please write here explicitly how the emissions are kept constant? Are they fixed to 1995 or the mean of the period etc: : :?

This sentence has been revised: "We also conducted a sensitivity simulation in which the anthropogenic emissions were kept constant (at the 1994 levels), to represent a scenario with fixed emissions throughout the years where observations are available to investigate the effects of emissions on ozone trends."

Line 190: The supplement figure should be referred here.

We have added the supplement figure (Fig. S6) reference in the revised sentence.

Line 231: the trends written in the text are slightly different than those on the plots, please double check.

We have modified the trends in the text.

Check the alphabetical order in the Referece list.

We have rearranged the reference list according to the alphabetical order.

Krotkov et al. (2016) is missing in the text.

We have added this reference in the text.

Langner et al. (2004) is missing in the reference list.

We have added Langner et al. (2004) in the reference list.

Change Lelieveld et al. (2000) with Lelieveld and Dentener (2010).

We have added the reference: Lawrence and Lelieveld (2010).

Fig. 1. It would be more interesting to see the e.g. annual mean O3 distribution rather than the surface elevation.

The revised Fig. 1 have added to show mean ozone mixing ratios for every five years during 1995-2014 over the selected 93 sites.

Legends should be added to all figures with time series plots.

We have added legends in the time series plots.

**Referee #2**

**General comments**

This paper by Yingying et al., investigates long-term trend in near-surface ozone in Europe by analysed observations part of the EMEP network. Moreover, it provides some very interesting hints about the different weights that change in European anthropogenic emission and "climate" variability have in determining the observed long-term tendencies.

The paper is well written and within the goal of ACP, the topic is more than relevant. Here, I addressed a few major and minor points that must be considered before final publication in ACP.

We thank the reviewer for comments, which have been incorporated to improve the manuscript.

**MAJOR POINTS**

1) One major point that must be carefully addressed by authors is the statistical significance of the tendencies reported in the paper. As an instance the Mann-Kendall test must be applied to the different subset of data to verify the actual existence of a "trend". Otherwise, the authors can only discuss about "tendencies". It is questionable to discuss and attribute tendencies that are not statistically significant, i.e. not different from zero. As an instance, statistical significance of tendencies/trends must be indicated in Table 3.

Thanks for the suggestion. In the revised manuscript, to address the statistical significance, all trends in the text and tables (Table 2 and Table 3) are performed with an F-test at the 95% confidence level.

2) By reading the paper is not clear to me how the authors aggregate data. Are the monthly percentiles (line 96-98) the average of the corresponding percentiles at each single station or the percentiles obtained for the whole data set (i.e. by considering all the ozone data observed at the 93 stations) for each specific month? I think the first "metric" would be much more robust that the second: : :

Yes, we calculate the monthly percentiles with the first method above to get the corresponding percentiles at individual station in each month. We analyze the ozone trends and variation for different percentiles at each station (Fig. 5, Fig.11, Fig. 12, and Fig. S2). Averaged over the 93 sites, we then also calculate the trends of different percentile ozone concentrations over the whole Europe. To make it clear, this sentence has been revised: "The monthly 5th, 50th and 95th percentile ozone concentrations for each period (per hour, daytime, nighttime and diurnal) are derived from the lowest, middle and highest 5th percentile hourly ozone mixing ratios of the corresponding period at individual stations in each month. Averaging over the 93 sites, we then also calculate the trends of different percentile ozone concentrations over the whole Europe."

3) The analysis concerning the impact of climate variability is promising but it need more attention: it is not novel that near-surface O3 respond to air-temperature (used as proxy of meteorological conditions favorable for photochemical production and accumulation). I would see a more deep discussion (and possibly analysis, see my comment about Fig S9) about the specific processes underlying this "climate variability". The authors mentioned (and reported by Figure S9) an influence of NAO but without any specific comments/explanation (I also suggest to discuss possible implication of NAO to air-mass transport regimes). As suggested by the Referee#1, biomass burning occurring at continental scale is an issue for near-surface ozone, especially under heat-wave or dry conditions. A cross-correlation analysis with number or geographical distribution (burned area) of open forest fire numbers can be useful to assess this point. For a large subset of year (i.e. since 2000), MODIS data can be used.

We have moved and revised the Fig. S9 to the main text and discussed more deeply in the revised Sect. 4.2.2:

"Fig. 11 shows the correlations between the monthly mean 2-meter temperature and the monthly mean, 5th and 95th percentile ozone for diurnal, daytime and nighttime concentrations. Most of these site-by-site correlations are statistically significant (P-value < 0.05 under a T-test; shown as triangles in Fig. 11) with high fraction (66%–91%) of sites for which significant correlation exist. For each metric (mean and percentiles for diurnal, daytime and nighttime), it corroborates the high correlations over central Europe with statistically significant values up to ~0.82 (P-value < 0.01). It indicates that the surface ozone mixing ratios are highly sensitive to enhanced air temperature, being favorable for photochemical O3 production, which has been reported by previous studies (Lin et al., 2017; Yan et al., 2018 and references therein). For different seasons, ozone variations in fall are most closely affected by temperature (Fig. S9), followed by the spring and summer ozone. The weakest linkage between ozone and temperature is in winter with few sites for which significant correlation exist especially for 95th percentile.

In contrast to the positive correlations over central and southern stations, ozone concentrations over the northern and western sites are negative and significantly correlated with temperature, associated with statistically insignificant correlations at several sites located in the transition regions from positive-correlation to negative-correlation (Fig. 11). This may be related to the influence of the Northern Atlantic Oscillation (NAO; a dominant mode of winter climate variability in the North Atlantic region including Europe; higher correlations with ozone in winter shown in Fig. S11), which had an opposite impact on ozone over northern and western compared to central and southern Europe (Fig. S10). This is because the positive

NAO phase is associated with enhanced pressure gradient between subtropical high pressure center (stronger than usual) and Icelandic low (deeper than normal). It can result in more and stronger winter storms crossing the Atlantic Ocean on a more northerly pathway, and consequently lead to warm and wet air in northern Europe. Compared to the impact of temperature, the effect of NAO on ozone are relatively modest with much lower correlations (Fig. 11 and Fig. S10). The correlations of less than 30% of the sites pass the significance test (P-value < 0.05). These results underscore that the large-scale climate variability affects the inter-annual variability of European background ozone."

Thanks for the suggestion of necessary discussion in biomass burning. Many previous studies have shown the linkage between forest fires under heat-wave and surface ozone. Here we have added this discussion in the revised Sect. 4.2.1:

"Especially, in August 2003, coinciding with a major heat wave in central and northern Europe, massive forest fires were observed from the Terra and MODIS satellite in many parts of Europe, particularly in the south and most pronounced in Portugal and Spain (Pace et al., 2005; Hodzic et al., 2006, 2007; Solberg et al., 2008). Long-range transport of fire emissions have been found to give rise to significantly elevated air pollution concentration and proved to be contributed to the European ozone peak values in August 2003 (Solberg et al., 2008; Tressol et al., 2008; Ordóñez et al., 2010)."

4) Line 394-395: the role of China emissions (even if reasonable) is not supported by data or analysis in this paper. If not strong evidences are added, this statement must be strongly understated or presented with much more caution. I'm wondering if you can use EMAC to make sensitivity study on China emission trends by playing with the MACCity inventory:

We agree with the referee that the sentence is not accurate. Although a sensitivity run can be performed with the EMAC model, we believe this to be out of the scope of this manuscript. Many studies have shown the impact of intercontinental transport on European ozone (i.e. Derwent et al. 2008, West et al. 2009a and West et al. 2009b). However, how claimed by the referee and shown by the aforementioned studies, emissions from other regions have larger impact on European ozone than Chinese emissions. Therefore we reformulated the sentence as: "Slower rates of ozone reduction during nighttime are suggested to be combined effects of reduced titration due to lower NOx emissions, and an increase in the global background ozone concentrations during this period, probably due to growing precursor emissions worldwide since 1995, which has been predicted by Lelieveld and Dentener (2000) based on atmospheric chemistry – transport modeling, and corroborated by satellite observations (Richter et al., 2005; Krotkov et al., 2016)."

We also revised the conclusions.

**SPECIFIC COMMENTS**

Line 71: annual "surface" 5th...maybe "surface" ozone concentration?

We have modified this sentence.

Line 96: please, better elucidate the aggregation process to obtain the calculated percentiles

Have revised; please see our response to major comment 2.

Line 170: which is the number sites characterized by negative trend ?

We have added this information: "For the ozone trend of 95th percentile at individual station, 84 sites (90%) are characterized by decreasing trend in daytime and 78 sites (84%) at night (Fig. 5 and Fig. S2)."

Line 187-190 and Table 3. Are these tendencies/trends obtained by averaging single trends/tendencies at each station or what else? Please specify.

These trends are calculated with the 5th, 50th, and 95th percentile ozone concentrations averaged over the 93 sites, not obtained by averaging trends at each station. We have specified in the revised text.

Line 202. Some comments are due to the absence of diurnal cycle for 5th percentile in winter. I would expect a diurnal cycle in NOx anthropogenic emissions that can affect O3 diurnal cycles and subsequently its trends: : :

Thanks for the suggestion. We have updated this sentence: "The slight growth rates in the 5th percentile ozone are approximately equally distributed at the level of  $0.1 \pm 0.12 \mu g/m^3/y$  (P-value > 0.05), probably due to the absence of ozone diurnal cycle, affected by NOx anthropogenic emissions, for 5th percentile especially in winter."

Line 214: did you calculate the average of trends or trend of averaged ozone over the whole Europe. In this latter case, you put together sites with very different inhomogeneous in term of ozone variability. As an instance, in summer, ozone is strongly dependent by geographical regions and latitudes: : :This is also evident by your Figure 4.

Here the trends are calculated with the averaged ozone over the whole Europe. To show the different inhomogeneous in term of ozone variability at different stations, we calculate the spatial standard deviation of trends:

$$\sigma = \sqrt{\frac{1}{N} \sum_{i=1}^{N} (\omega_i - \alpha)^2}$$

where N is the total number of sites,  $\omega_i$  is ozone trend at individual sites and  $\alpha$  represents the average ozone trend.

The ozone trends at each station and the average of trends are also show in Fig.5 and Fig. S2.

Line 233: the annual trend for emep stations here reported are different from those in Table 3. Why?

Here the annual ozone trends (Fig. 5) are the average of trends at each station. The trends in Table 3 are calculated with the averaged ozone over the 93 sites.

Line 234-235:please comment these geographical differences and provide possible reasons

We have added the possible reasons: "These geographical differences in ozone trends are probably explained by the effects of a general decrease in European anthropogenic precursor emissions, being partly offset by those of climate variability (see Sect. 4.2 for discussion of Fig. 11 and Fig. S10)."

Line 237: what do you mean with "regional trend contrast". Contrast in respect to what?

Here "regional trend contrast" means the geographical differences in ozone trends. We have updated this sentence: "The geographical differences in ozone trends are most significant in spring with an average growth rate of 0.01  $\mu$ g/m3/y (Fig. 5)."

Line 251: why did you investigate the correlation with 95th percentile? What do you want to proof?

We calculate the correlation between the exceedances and 95th percentile ozone to show their interannual consistence.

Line 275: is the trend overestimation (especially for 95th percentile, i.e. lower decrease with time) due to the O3 overestimation since 2010?

Thanks for the suggestion. Yes, the ozone overestimation since 2010 may be the dominant reason for the trend overestimation. We have added this comment in the revised text.

Line 297: what the reason of the enhanced trends in the 5th percentiles?

We have added the possible reason: "The possible reason for these simulated enhanced ozone trends is the overestimation of the decline of European anthropogenic ozone precursor emissions (decreasing more rapidly than observed) in EMAC."

Line 359: Figure S9 need to be shown in the main body of paper and it deserve more attention/comments/explanation. As an instance, what the possible impact of NAO variability to transport regimes?

We have moved this figure to the main text and please see our response to major comment 3 for detail discussion.

Figure S8-S9: please identify the sites with statistically significant correlation and provide in the paper the fraction of sites for which significant correlation exist for each metric (mean, percentiles) with T and NAO.

In the revised Fig.11 and Fig. S10, we have identified the sites with statistically

significant correlation and also shown the fraction of sites for which significant correlation exist for each metric (mean, percentiles) with T and NAO in the figures. We have also discussed it in the text; please see our response to major comment 3.

Line 352: are these correlation calculated over the 20-yr period? Since NAO effect are strongly dependent by season (see Pausata et al., ACP, 2012), Fig S8 and S9 should be disaggregated as a function of different seasons.

Yes, these correlations are calculated over 1995-2014. We have shown the seasonal correlations in the revised Fig. S9 and Fig. S11 and also added some discussion in the revised Sect. 4.2.2.

Line 379: it may be useful if the fraction of sites with statistically significant trends is provided.

Have added: "Results show that although reductions in anthropogenic emissions have lowered the peak ozone concentrations (sites with statistically significant trends: 91 out of 93 sites; 98%), especially during daytime in the period 1995–2014, the lower level ozone concentrations have increased (sites with statistically significant trends: 71 out of 93 sites; 76%) continually since 1995 over Europe."

In the "Conclusion section" it should be stressed that 20-yr is a time frame too short for depict climate tendency (formally a 30 yr period is necessary). I agree that some "large-scale" processes like NAO can influence near-surface O3, thus possible change of these regimes under a changing climate can have serious impact on ozone.

We have added this discussion in the revised conclusion: "We note that our analysis over 1995–2014 is a timeframe too short for the analysis of climate tendencies (formally a 30-year period is necessary). Thus, here the climate related variability is mainly driven by the large-scale processes like NAO and heat wave occurrence, which may be influenced by climate change."

Reference:

Derwent, R. G., et al. "How is surface ozone in Europe linked to Asian and North American NOx emissions?." Atmospheric Environment 42.32 (2008): 7412-7422.

West, J. Jason, et al. "Effect of regional precursor emission controls on long-range ozone transport–Part 1: Short-term changes in ozone air quality." Atmospheric Chemistry and Physics 9.16 (2009): 6077-6093.

West, J. Jason, et al. "Effect of regional precursor emission controls on long-range ozone transport–Part 2: Steady-state changes in ozone air quality and impacts on human mortality." Atmospheric Chemistry and Physics 9.16 (2009): 6095-6107.